EMBO
Molecular Medicine

# Gain-of-function mutation in TASK-4 channels and severe cardiac conduction disorder

Corinna Friedrich[1,†], Susanne Rinné[2,†], Sven Zumhagen[1], Aytug K Kiper[2], Nicole Silbernagel[2], Michael F Netter[2], Birgit Stallmeyer[1], Eric Schulze-Bahr[1,‡] & Niels Decher[2*,‡]

## Abstract

Analyzing a patient with progressive and severe cardiac conduction disorder combined with idiopathic ventricular fibrillation (IVF), we identified a splice site mutation in the sodium channel gene *SCN5A*. Due to the severe phenotype, we performed whole-exome sequencing (WES) and identified an additional mutation in the *KCNK17* gene encoding the K$_{2P}$ potassium channel TASK-4. The heterozygous change (c.262G>A) resulted in the p.Gly88Arg mutation in the first extracellular pore loop. Mutant TASK-4 channels generated threefold increased currents, while surface expression was unchanged, indicating enhanced conductivity. When co-expressed with wild-type channels, the gain-of-function by G88R was conferred in a dominant-active manner. We demonstrate that *KCNK17* is strongly expressed in human Purkinje cells and that overexpression of G88R leads to a hyperpolarization and strong slowing of the upstroke velocity of spontaneously beating HL-1 cells. Thus, we propose that a gain-of-function by TASK-4 in the conduction system might aggravate slowed conductivity by the loss of sodium channel function. Moreover, WES supports a second hit-hypothesis in severe arrhythmia cases and identified *KCNK17* as a novel arrhythmia gene.

**Keywords** arrhythmia; K$_{2P}$ channels; progressive cardiac conduction disorder; SCN5A
**Subject Categories** Cardiovascular System; Genetics, Gene Therapy & Genetic Disease

## Introduction

Progressive cardiac conduction disorder (PCCD) is a common disease affecting primarily patients in the fifth decade of life. The hallmark of the disease is a progressive slowing of cardiac conduction, affecting the His-Purkinje system with left or right bundle branch block (LBBB/RBBB) and prolongation of the QRS complex. PCCD is one of the main causes of pacemaker implantations. The genetic basis for PCCD mostly remains elusive, albeit in 20–30% of probands, disease-causing mutations were reported in several genes (Fatkin *et al*, 1999; Schott *et al*, 1999; Watanabe *et al*, 2008; Kruse *et al*, 2009). A splice site mutation in *SCN5A*, leading to a deletion in the cardiac sodium channel Nav1.5, was the first reported genetic cause of progressive or non-progressive CCD (Schott *et al*, 1999). Loss-of-function mutations in the beta subunit of Nav1.5 (*SCN1B*) lead to PCCD as well (Watanabe *et al*, 2008). Furthermore, gain-of-function mutations in the transient receptor potential cation channel gene *TRPM4* in a large South African pedigree with an autosomal dominant form of PCCD were described as an important cause for PCCD (Kruse *et al*, 2009; Stallmeyer *et al*, 2012). In addition, mutations in *LMNA* that encodes for the structural protein lamin A/C of the nuclear lamina are associated with dilated cardiomyopathy (DCM) and conduction system disease (CMD1A) (Fatkin *et al*, 1999).

Idiopathic ventricular fibrillation (IVF) is characterized by spontaneous ventricular fibrillation in the absence of structural and primary electrical heart diseases. In addition, overlap features with other electrical heart diseases such as early repolarization syndrome or Brugada syndrome (BrS) are known. In contrast to PCCD, patients with IVF are in need of an implantable cardioverter defibrillator (ICD). Unfortunately, the identification of genes involved in IVF is difficult. Classical linkage analysis is hampered since the diagnosis of the disorder cannot be made on the basis of electrocardiography (ECG) recordings, but only in presence of a life-threatening VF event in absence of other causes. In addition, many affected patients die young, and their DNA is mostly not available for genetic analysis. Several genes have been associated with IVF. Some loss-of-function mutations in the cardiac sodium channel gene *SCN5A* cause IVF in the absence of the classical BrS phenotypes (Akai *et al*, 2000). Mutations in the *SCN3B* sodium channel subunit (Navβ3) also result in a decreased sodium current and IVF (Valdivia *et al*, 2010). Linkage of the *DPP6* locus (encoding dipeptidyl-peptidase 6, which modulates the transient outward current ($I_{to}$) kinetics) was reported in a familial IVF pedigree (Alders *et al*, 2009). In addition,

1   Department of Cardiovascular Medicine, Institute for Genetics of Heart Diseases (IfGH), University Hospital Münster, Münster, Germany
2   Institute of Physiology and Pathophysiology, Vegetative Physiology, University of Marburg, Marburg, Germany
   *Corresponding author. Tel.: +49 6421 28 62148; Fax: +49 6421 28 66659; E-mail: decher@staff.uni-marburg.de
   †Susanne Rinné and Corinna Friedrich contributed equally to this work.
   ‡Eric Schulze-Bahr and Niels Decher are co-senior authors.

a mutation in the *KCNJ8*-encoded pore-forming subunit of the ATP-sensitive potassium channel Kir6.1 was found in single cases with IVF; however, the functional effect of this mutation is not fully understood so far (Haissaguerre *et al*, 2009).

Conventional genotyping for studying cardiac arrhythmia of unclear genesis and the impact of sporadic genetic variation on heart function requires the knowledge of underlying genes. In the last years, next-generation sequencing (NGS) technologies have been emerged and exome sequencing data have been rapidly provided to identify novel disease causes (Ng *et al*, 2008; de Ligt *et al*, 2012; Crotti *et al*, 2013). Particularly for Mendelian disorders, whole-exome sequencing (WES) evolved as a suitable approach for disease gene identification since it offers to achieve sequence information of nearly all coding regions that are expected to harbor the majority of mutations, at once. In this context, it is noteworthy that recent studies on patients with inherited forms of arrhythmias reported on a combination of distinct mutations and additional, polymorphic variants to exaggerate the clinical phenotype. As an example, a mutation in *SCN5A*—in conjunction with two rare polymorphisms in regulatory regions of the gene encoding connexin40 (*CX40*)—is the genetic cause of familial atrial standstill and family members with either of the genotypes were clinically unaffected (Groenewegen *et al*, 2003). In congenital long-QT syndrome (LQTS), a common single nucleotide polymorphism (SNP) in *KCNH2* (p.Lys897Thr) is proposed to act as a genetic modifier in presence of the p.Ala1116Val in the same gene. Bearing solely one of those variants, patients are asymptomatic or only have a latent form of LQTS, respectively (Crotti *et al*, 2005). Not only SNPs can modify a LQTS phenotype, but patients being genotyped as compound heterozygous for mutations in different LQT genes present a more severe phenotype than patients with only one mutation (Millat *et al*, 2006). Similarly, for idiopathic epilepsy, it has been described that it is not only based on the presence of one particular SNP in an ion channel gene, but the interplay of different genetic variants which determine the personal risk assessment for this idiopathic neurological disorder (Klassen *et al*, 2011).

In the present study, we investigated the genetic causes of arrhythmias in a patient with a severe cardiac phenotype combined of PCCD and IVF. After identifying a splice site mutation in *SCN5A* using the conventional Sanger sequencing technique, we thought to reanalyze this particular case in light of the pronounced phenotype and applied WES to search for additional genetic causes and, thus, for other disease genes or modifiers. Following a prioritization scheme for identified variants, we finally discovered an additional, relevant mutation in the *KCNK17* gene, encoding the pH-sensitive cardiac two-pore domain potassium channel ($K_{2P}$) TASK-4 (Decher *et al*, 2001), also known as TALK-2 (Girard *et al*, 2001). This is the first description of a $K_{2P}$ channel linked to an inherited form of arrhythmia. Functionally, the identified non-synonymous mutation p.Gly88Arg that is located in the first extracellular loop of TASK-4 resulted in threefold increased current amplitudes by an altered gating behavior which was conferred to wild-type channels in a dominant-active manner. This gain-of-channel function might lead to reentry arrhythmias due to an enhanced rate of ventricular repolarization. As *KCNK17*/TASK-4 transcripts have been found more pronounced in human Purkinje fibers, the gain-of-function due to G88R might cause a hyperpolarization of cells in the conduction

system and slowed conductivity by aggravating the effects of the *SCN5A* splice site mutant.

Concluding, in the present case, the arrhythmia phenotype of PCCD with IVF is not solely based on the presence of a single, particular mutation, but more likely a result of the interaction of multiple genes and mutations. Therefore, WES might be a powerful clinical tool to identify multiple causes for distinct and severe cardiac phenotypes. We have shown here the first mutation in a $K_{2P}$ channel acting as a modifier of human cardiac arrhythmia providing evidence that $K_{2P}$ channels are important regulators of cardiac excitability.

# Results

## Case presentation

A 63-year-old man with recurrent syncope and non-sustained ventricular tachycardia (VT; left bundle branch morphology, cycle length 315 ms) was investigated in our institution. The ECG at rest showed an atrioventricular (AV) block I° (PQ interval: 205 ms) as well as a right bundle branch block (QRS interval: 115 ms; RBBB) (Fig 1A) which was progressive within a time frame of 5 years (PQ interval: 211 ms, QRS interval: 166 ms) (Fig 1B). A coronary angiography and cardiac magnetic resonance imaging were normal. An invasive electrophysiological study identified a prolonged atrio-His (AH) interval of 80 ms, a frequency-dependent left bundle branch block (LBBB) and an increased atrial vulnerability. In addition, programmed ventricular stimulation (500 ms, S3 pacing) at the right ventricular (RV) apex induced VF (but not a monomorphic VT), being externally defibrillated and terminated. Subsequently, an ICD has been implanted. During follow-up, several episodes of VT could be effectively terminated by the ICD. The family history was negative for sudden cardiac death or known inherited cardiac conditions.

## Candidate gene sequencing and WES

After sequencing the key genes (e.g., *TRPM4, LMNA*) for PCCD, we first identified in the *SCN5A* gene an unpublished, single nucleotide exchange in the essential donor splice site of intron 22 (c.3963+1G>A) (Fig 2A). The boundary between exon 22 and exon 23 is located in the intracellular linker between transmembrane segments S4 and S5 in domain 3 (DIII) of the Nav1.5 α-subunit (Fig 2A–C). In consequence, skipping of exon 22 is very likely, since a mutation affecting the same splice site (c.3963+2T>C, reported as IVS22+2T>C) was shown to cause exonic skipping and a complete loss of channel function (Schott *et al*, 1999).

Because of the unusual and severe phenotype (namely PCCD and IVF), additionally WES was applied to identify concomitant genetic causes. A total of 9.94 Gb of sequence data were read, and 99.39% were aligned at high quality to the reference genome (hg19 build) (Table 1). The median read depth was 107.9-fold which is remarkably higher than the estimated average depth (33-fold) required for highly accurate downstream heterozygous variant detection (Xu *et al*, 2011). A minimum of 95% of the on-target regions were covered to a depth of at least 20-fold and, thereby, considered for variant filtering to ensure good detection sensitivity; only 0.1% of the exome sequences were not covered (Table 1).

**Table 1. Overview of whole exome sequencing data in a proband with PCCD and IVF**

| | Exome data analysis |
|---|---|
| Total sequence data (Gb) | 9.94 |
| Reads aligned (%) | 99.39 |
| Total number of reads aligned in pairs | 103,717,004 |
| Median coverage (range: 0–250-fold) | 107.9-fold |
| 0-fold coverage (%) | 0.1 |
| >20-fold coverage (%) | 95.00 |

Gb, Giga basepairs.

## Nucleotide variant analysis of the whole exome and variant selection

Following a prioritization scheme (Fig 2D), only nucleotide variants with a minimum coverage of 20× were included (i.e., 76,712 nucleotide variants in 19,135 genes). Among them, 1,589 (2.1%) were present in an in-house priority list (CARDIO panel) of 388 relevant genes (2.0% of all genes) for cardiac function and/or inherited cardiac conditions. In 110 of the 388 genes, no nucleotide variant was detected, respectively.

Next, in the 278 remaining priority genes, nucleotide variants without serious amino acid consequences (*n* = 1,500 variants) were

excluded. Out of 89 (5.6%) remaining nucleotide variants, seven were absent or very rare (i.e., MAF < 0.01%) in common genomic databases (e.g., EVS, 1,000 genomes, dbSNP, Ensembl Gene Browser), whereas 82 were already known and not further considered (Supplementary Table S1). In Table 2, the remaining seven nucleotide variants are listed: among them, four were predicted to cause a non-synonymous (ns) amino acid exchange, one an in-frame deletion, one an insertion, and finally, one an essential splice site alteration. Each nucleotide variant was independently confirmed by direct sequencing of both strands in Sanger technique.

The pathogenic impact of the so confirmed ns single nucleotide variants (nsSNV) was determined using five different *in silico* pathogenicity prediction tools (PPT) (Table 2). Two nsSNV, in the chloride channel gene *CLCN7* (p.Pro71Leu) and *RAF1* gene (p.Asp203Asn), were concordantly described as 'tolerated' and therefore considered as rare, but not disease-causative nsSNP. Mutations in *RAF1,* which encodes a protein kinase involved in MAP kinase pathway, cause Noonan syndrome 5 and Leopard syndrome 2 which are completely unrelated to the present cardiac phenotype seen here. Another amino acid substitution (in *FLNC* gene: p.Val749Ala) had a discrepant pathogenicity prediction (2/5: 'tolerated' upon program prediction) and was consequently denoted as variant of uncertain significance (VUS) in the absence of functional evidence. *FLNC* codes for filamin C, a structural protein being also expressed in cardiac tissue. However, *FLNC* mutations are associated with myop-

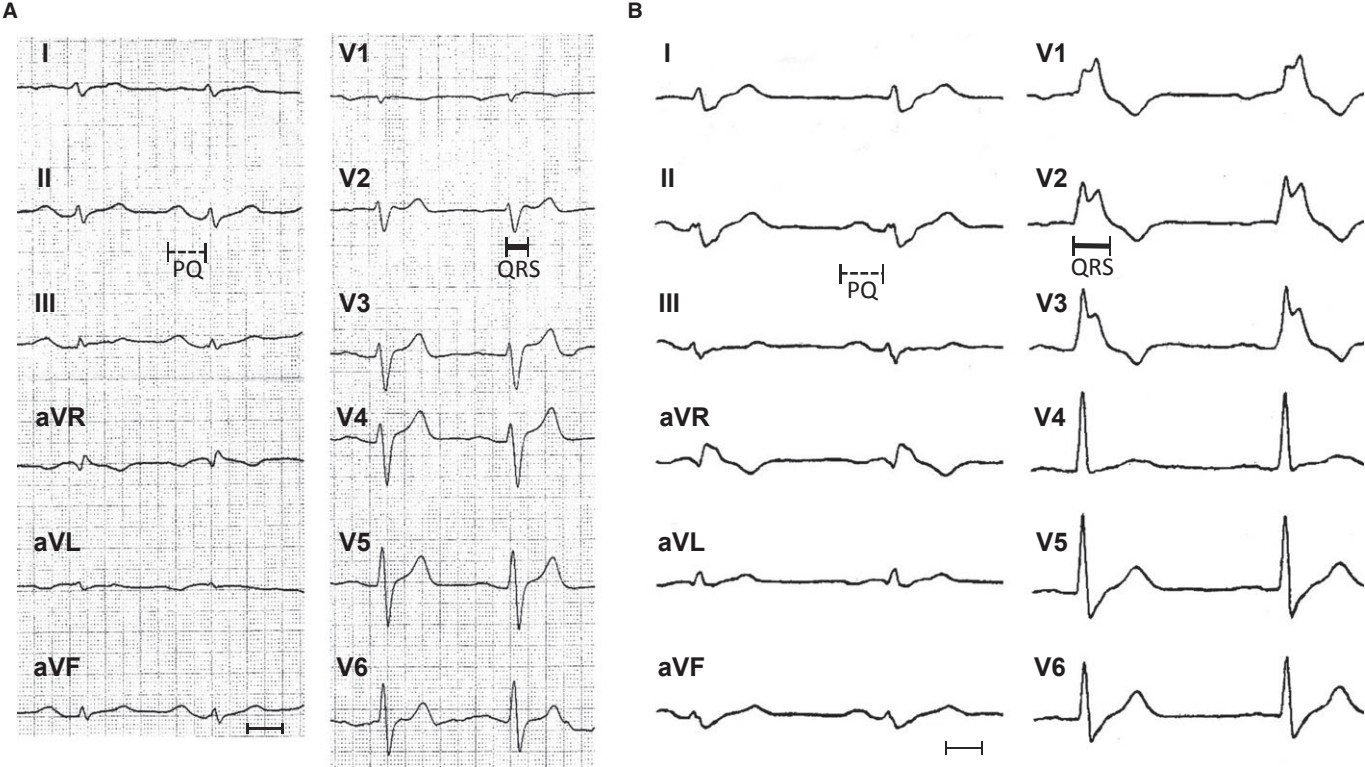

**Figure 1. Twelve-lead electrocardiography (ECG) of a patient with progressive cardiac conduction disorder (PCCD) and idiopathic ventricular fibrillation (IVF).**

A    In 2003, heart rate (HR) 79 bpm, PQ interval 205 ms (marked by a dotted line), QRS 115 ms (continuous line), QT 351 ms, and corrected QT interval 402 ms.

B    In 2008, HR 59 bpm, PQ interval 211 ms (dotted), QRS 166 ms (continuous), QT 433 ms, and QTc interval 427 ms (measurements each in lead II; paper speed 50 mm/s, scale bar 10 mm).

**Table 2.  Nucleotide variants with serious consequences identified by WES in a patient with cardiac conduction defect with IVF: results of pathogenicity prediction**

| Gene | NCBI accession no. | cDNA Level | Protein level | Pathogenicity prediction | Annotation |
|---|---|---|---|---|---|
| Non-synonymous variants[a] | | | | | |
| CLCN7 | NM_001287.5 | c.212C>T | p.Pro71Leu | Tolerated (100%)[b] | rs145267254 (0.008%) |
| FLNC | NM_001458.4 | c.2245G>A | p.Val749Ala | Discordant (40%)[c] | novel, unpublished |
| KCNK17 | NM_031460.3 | c.262G>A | p.Gly88Arg | Damaging (100%)[b] | rs141016843 (0.008%) |
| RAF1 | NM_002880.3 | c.607G>A | p.Asp203Asn | Tolerated (100%)[b] | novel, unpublished |
| Insertion/deletion variants | | | | | |
| CLCA4[d] | NM_012128.3 | c.2656_2661del | p.Thr886_Pro887del | Tolerated | rs67478973 |
| SMAD5[e] | NM_001001419.1 | c.1314_1315insC | Not altered | Tolerated | rs55765823 (0.000%) |
| Splice site variants[e] | | | | | |
| SCN5A | NM_198056.2 | c.3963+1G>A | Skip of exon 22 | Damaging | novel, unpublished |

[a]Pathogenicity amended by PolyPhen-2, SIFT, MutPred, SNPs&Go, SNAP.
[b]100% of pathogenicity prediction tools (PPT) predict the same effect.
[c]40% of PPT predict neutral impact.
[d]Pathogenicity amended by SIFT/Provean.
[e]Pathogenicity amended by Alamut.

athy with cardiac involvement or with Limb-Girdle dystrophy, a mainly skeletal muscle disorder. The six base pairs in-frame deletion in *CLCA4* (resulting in the two amino acid deletion p.Thr886_Pro887del) was evaluated as 'tolerated' and therefore also not further considered. *CLCA4* encodes for the calcium-activated chloride channel regulator 4 that is primarily expressed in the digestive system and to a lower extent in cardiac tissue. Mutations in *CLCA4* can cause cystic fibrosis not being present in the patient. The insertion in the *SMAD5* gene (c.1314_1315insC) was not further analyzed since it did not alter protein coding upon *in silico* analysis (Alamut program). *SMAD5* codes for SMA- and MAD-related protein 5 involved in TGFβ signaling. It is expressed in cardiac tissue and is indirectly associated with arterial pulmonary hypertension that, however, was not seen in transthoracic echocardiography in the patient.

While typical genes causing IVF (e.g., *SCN3B, KCNJ8, KCNQ1, RYR2, TNNT2, CACNA1C, CACNB2, CACNA2D1, DPP6*) or PCCD (e.g., *TRPM4, LMNA*) were without any detected variant, WES again identified and thereby confirmed the intronic nucleotide exchange in the *SCN5A* gene (c.3963+1G>A), as noted before. Another splice site mutation affecting the same essential splice site (published as IVS 22+2T>C, corresponding to c.3963+2T>C) is known to result in skipping of exon 22 and a loss of channel function in whole-cell patch clamp recordings (Probst *et al*, 2003). Loss-of-function mutations in the cardiac sodium channel gene *SCN5A* are associated with BrS (Kapplinger *et al*, 2010), PCCD (3), DCM in combination with atrial and ventricular arrhythmias and conduction disease (McNair *et al*, 2004), sick sinus syndrome (SSS) (Benson *et al*, 2003), familial atrial fibrillation (AF) (Darbar *et al*, 2008), and sudden cardiac infant death syndrome (SIDS) (Klaver *et al*, 2011).

In addition, apart from the intron mutation in *SCN5A,* one of the seven amino acid variants was found in the *KCNK17* gene encoding the TASK-4 channel belonging to the $K_{2P}$ potassium channel family. All PPTs concomitantly evaluated p.Gly88Arg (shortly: G88R) in the *KCNK17* gene as 'damaging.' The heterozygous single base pair exchange (c.262G>A) results in a glycine to arginine amino acid exchange at position 88 in the first extracellular loop of TASK-4

(Fig 2E and F). The glycine residue at this site is highly conserved throughout orthologous channels (Fig 2G). Taken together, starting with 76,712 variants in 19,135 genes prioritization of variants filtered out the known *SCN5A* (c.3963+1G>A, p.?) single nucleotide substitution and the putative novel disease-modifying exchange in *KCNK17* (c.262G>A, p.Gly88Arg). All known family members were contacted and refused genetic testing. Since DNA from other family members was not available, we were not able to proof whether the identified genetic mutations in both genes were inherited or occurred as *de novo* ones (Supplementary Fig S1); however, the family history was not further indicative for other arrhythmias or sudden cardiac death.

### *KCNK17* genotyping of other patient cohorts and controls

In 463 patients with selected arrhythmia syndromes, including AF ($n = 10$), atrioventricular block (AVB) ($n = 20$), BrS ($n = 200$), IVF ($n = 125$), PCCD ($n = 49$), right ventricular outflow tract tachycardia (RVOT, $n = 35$), and sinus node disease (SND, $n = 24$), we completely sequenced all coding exons and adjacent intronic sites of *KCNK17*. Here, we identified a total of 14 polymorphic nucleotide sites (Supplementary Fig S2); these were not further considered as disease-causing because of a benign pathogenicity prediction. In addition, the G88R mutation did not appear in another patient and was also absent in 379 unrelated, healthy controls of the same ethnicity.

### *KCNK17*/TASK-4 expression in human heart compartments

Utilizing quantitative PCR experiments, we found that TASK-4 has the strongest expression in Purkinje fibers and the AV-node, followed by expression in the atria and sinoatrial (SA)-node (Fig 3A). TASK-4 has a strongly increased relative expression in AV-node (about sixfold) and Purkinje fibers (about 12-fold) compared to the ventricles (Fig 3A). The expression of TASK-4 in Purkinje fibers was as high as for the $K_{2P}$ channel TASK-1 (*KCNK3*), which is known to be preferentially expressed in the conduction system of mouse and chick hearts (Fig 3B) (Graham *et al*, 2006).

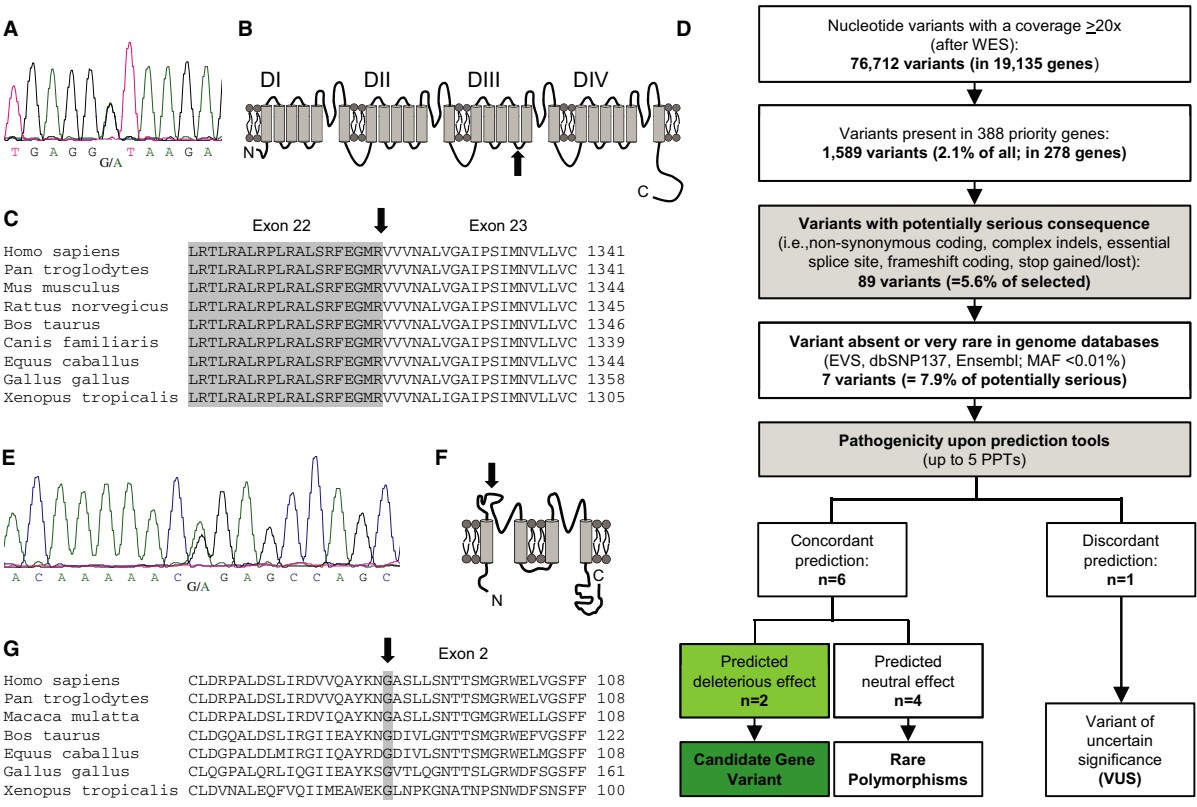

**Figure 2.  Heterozygous, digenic mutation state (*KCNK17* and *SCN5A*) in patient 10192-1.**

A  Electropherogram after direct sequencing of *SCN5*A: in the essential donor splice site located in intron 22, a heterozygous nucleic acid substitution was identified (c.3963+1G>A).

B  Schematic topology of the Nav1.5 α-subunit. The affected splice site (black arrow) is located in the intracellular linker between the transmembrane segments S4 and S5 of domain 3 (DIII).

C  Alignment of the boundary of exon 22/exon 23, position of the mutation is indicated by a black arrow. A skipping of exon 22 is predicted.

D  Prioritization scheme for filtering variants obtained by whole-exome sequencing (WES). Minimum read depth was 20×. First, filtering of variants for relevant heart genes was done. Next, all variants with non-serious consequences (synonymous and mostly intronic changes) were excluded. Only unknown or rare alterations (MAF < 0.01%) were further evaluated with pathogenicity prediction programs (nsSNV: PolyPhen-2, SIFT, MutPred, SNPs&Go, SNAP; in-frame indels: SIFT/Provean; frameshift, splice site variants: Alamut). If all programs concordantly predict a damaging effect, the related gene was classified as a candidate gene. Discrepant prediction results lead to a classification as a variant of uncertain significance (VUS).

E  Electropherogram after direct sequencing of *KCNK17*. A heterozygous nucleic acid substitution (c.262G>A) was detected in exon 2 of the *KCNK17* gene.

F  Schematic topology of the TASK-4 α-subunit with the point mutation p.Gly88Arg located in the extracellular linker between the first two transmembrane domains.

G  Alignment illustrating the high degree of conservation of G88 between orthologs of TASK-4.

## The G88R mutation causes a gain-of-function of TASK-4 currents

To study the functional consequence and explore the disease-causing/disease-modifying mechanism of the G88R exchange, we heterologously expressed the G88R mutant TASK-4 channels in *Xenopus laevis* oocytes and recorded current voltage relationships of wild-type and mutant TASK-4 channels (Fig 4A). Strikingly, mutant TASK-4 channels (G88R) generated approximately threefold more outward currents than wild-type channels (Fig 4A and B). The normalized TASK-4 currents increased from $1.0 \pm 0.04$ ($n = 71$) for wild-type to $2.93 \pm 0.13$ ($n = 70$) for G88R.

## Similar cell surface expression of mutant (G88R) and wild-type TASK-4 channels indicates an altered gating of G88R channels

The observed increase in current amplitude of the G88R mutant might be due to an increased surface expression or an altered gating

behavior of the channel. In order to quantify the surface expression of the G88R mutant, we introduced an extracellular HA epitope in the P2-M4 linker of TASK-4. Surface expression was subsequently analyzed with a chemiluminometric assay, as previously described (Zerangue *et al*, 1999; Decher *et al*, 2004). The G88R mutant showed a similar surface expression as wild-type TASK-4 channels (Fig 4C). Next, live-cell imaging of EGFP-tagged TASK-4 and DsRed-tagged G88R channels in HeLa cells was used to test whether the mutant channel has a changed cellular distribution pattern or surface expression in a mammalian cell line. EGFP and DsRed fluorescence are depicted in green and magenta, so that co-localization merges to white. After transfection of wild-type TASK-4-EGFP and DsRed-G88R channels, a similar expression pattern was observed over a time period of 12–72 h. Figure 4D illustrates cells 22 h after co-transfection with wild-type TASK-4-EGFP and DsRed-G88R channels. Wild-type and mutant channels co-localized throughout the cell and at the surface membrane (Fig 4D). Thus, the gain-of-

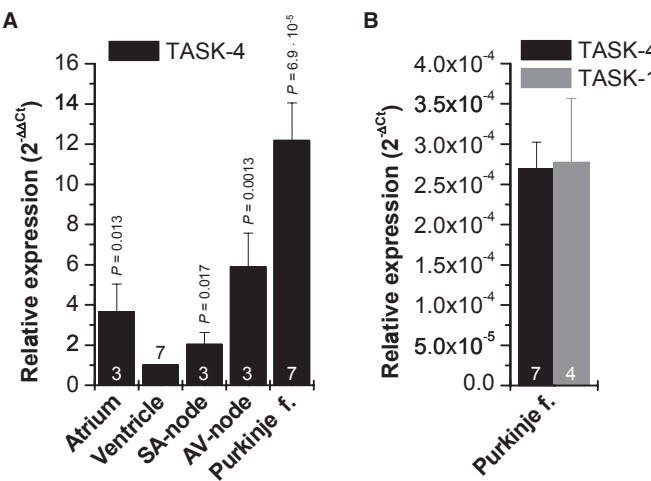

**Figure 3.  Expression pattern of TASK-4 in human cardiac compartments after quantitative RT-PCR.**

A   Relative values normalized to ventricular expression. f., fibers. The relative expression was calculated by dividing means of Ct-values of each sample by mean Ct of ventricle and setting into $2^{-\Delta\Delta Ct}$. The experiments were performed as triplicates and averaged from three to seven repeats. Statistical significance is indicated compared to ventricular *KCNK17* expression.

B   Ct-values for *KCNK17* (TASK-4) compared to *KCNK3* (TASK-1) in Purkinje fibers after normalization for *GAPDH*.

Data are provided as mean ± SEM. *P* values calculated in unpaired Student's *t*-test are indicated. Numbers of independent experiments are indicated within the bars.

function by the G88R mutation is not caused by an altered surface expression and accordingly must be due to an altered channel function/gating.

**Glycine 88 is a key residue for TASK-4 channel gating**

As described above, the gain-of-function of the G88R mutant must be caused by an altered channel gating which is consistent with the localization of the mutation near the selectivity filter (Fig 5A). TASK-4 is a pH-sensitive channel exhibiting an increased open probability upon extracellular alkalization, resulting in more outward currents (Decher *et al*, 2001). Thus, we tested whether the gain-of-function by G88R is caused by an altered pH sensitivity of TASK-4 channels. However, the G88R mutants showed similar activation by extracellular alkalization as wild-type channels (Fig 5B). Only at a pH of 10.5, G88R mutant channels showed less current increase than wild-type channels (*P* = 0.041) which was, however, not pronounced. Thus, the gain-of-function by G88R appears to be not primarily caused by an altered sensitivity of the TASK-4 channels to extracellular protons.

To analyze whether the glycine to arginine exchange in the extracellular M1-P1 linker causes an increased open probability of TASK-4 channels via the charge introduced at position 88, we tested whether the gain-of-function was also present in G88K or G88E mutants. Despite the differences in the charge introduced at this site, both mutations resulted in a gain-of-function (Fig 5C). G88K channels generated 7.3-fold more and G88E mutants 3.6-fold more outward currents than wild-type channels. Thus, we excluded a gain-of-function by a simple charge effect as a mechanism, as an

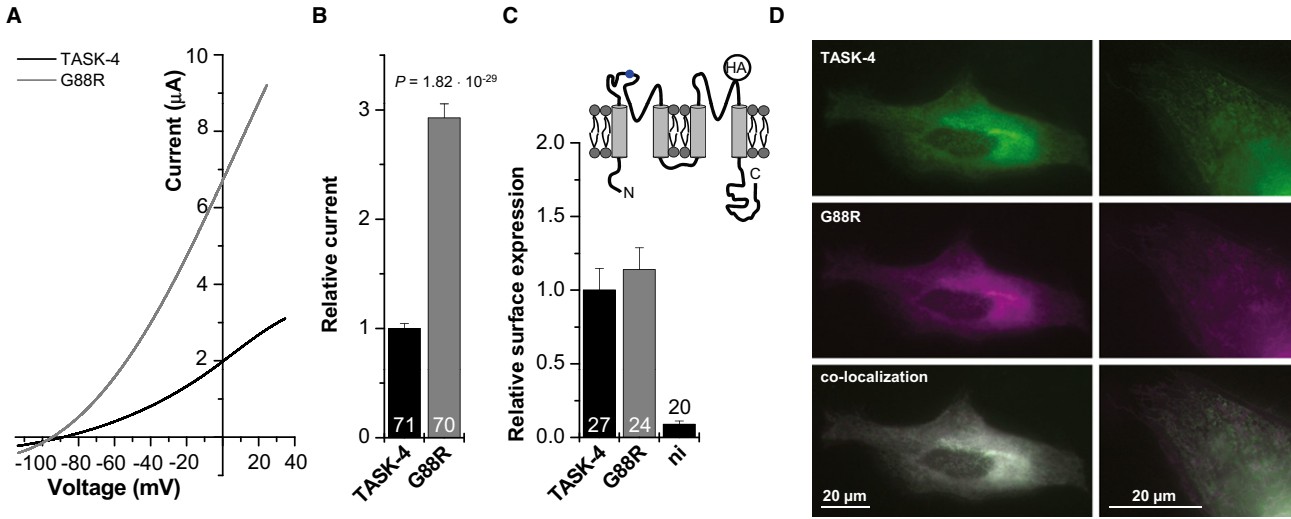

**Figure 4.  The p.G88R mutation in TASK-4 results in a gain-of-function in presence of a normal cellular distribution pattern and cell surface expression.**

A   Representative two-electrode voltage-clamp measurements in *Xenopus* oocytes injected with wild-type (black) or mutant (p.G88R, gray) TASK-4 cRNA (25 ng/oocyte). Voltage was ramped from −110 to +35 mV within 3.5 s. Holding potential was −80 mV and voltage ramps were repeated every 4 s.

B   Mean current amplitudes of wild-type- and G88R-TASK-4-mediated currents were analyzed at 0 mV from six independent sets of experiments. Relative current: TASK-4 = 1.0 ± 0.04 (*n* = 71) and G88R-TASK-4 = 2.93 ± 0.13 (*n* = 70).

C   Luminometric quantification of the surface expression of HA-tagged TASK-4 constructs. Relative surface expression normalized to wild-type TASK-4 is plotted. ni: non-injected oocytes. TASK-4 = 1.0 ± 0.15 (*n* = 27); G88R-TASK-4 = 1.14 ± 0.15 (*n* = 24); ni = 0.09 ± 0.02 (*n* = 20). The inset illustrates the membrane topology of a TASK-4 α-subunit and the localization of the HA-tag introduced in the P2-M4 linker. The position of the G88R mutation is indicated in blue.

D   Live-cell imaging of HeLa cells 22 h after transfection with EGFP-tagged wild-type and DsRed-tagged G88R-TASK-4 channels.

Data are provided as mean ± SEM. *P* values calculated in unpaired Student's *t*-test are indicated. Numbers of independent experiments are indicated within the bars.

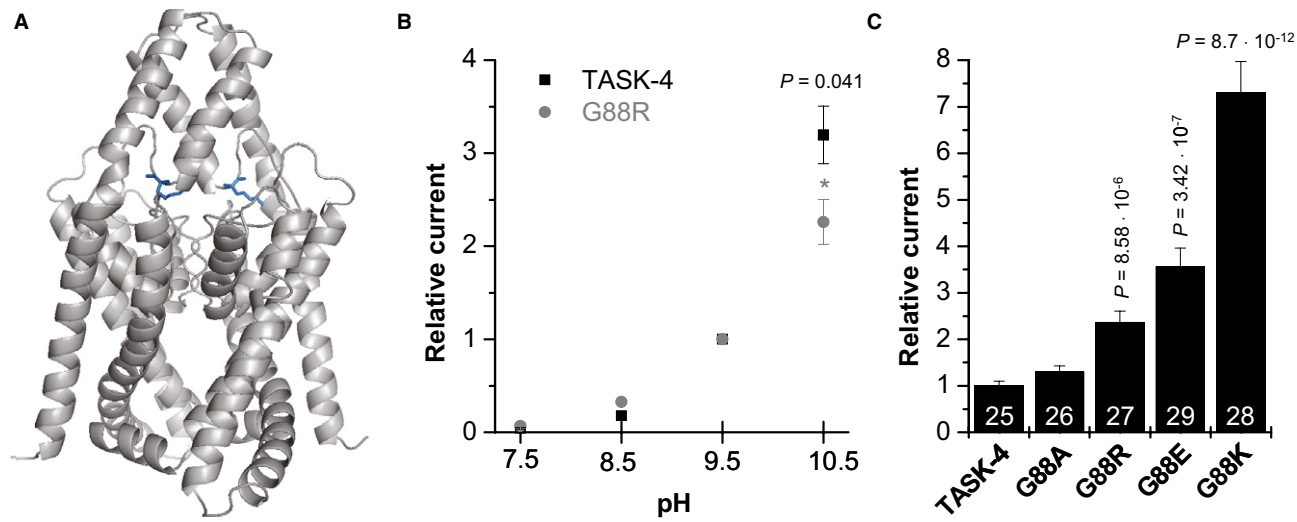

**Figure 5.  Glycine residue 88 is critical for the gating of TASK-4 channels.**

A   Localization of the G88 residue in a TASK-4 pore homology model based on the crystal structure of TWIK-1. The arginine (R) residues within the dimer are illustrated in blue.

B   Wild-type TASK-4 and G88R-TASK-4 channel sensitivity to external pH. Mean current amplitudes were recorded by 1-s test pulses to +40 mV, repeated every 4 s. Holding potential was −80 mV. Current amplitudes were analyzed at the end of the test pulse, with extracellular pH values ranging from 7.5 to 10.5. Data were normalized to the currents recorded at pH 9.5 and illustrated as mean ± SEM.

C   Mean current amplitude of oocytes injected with wild-type TASK-4 cRNA ($n = 25$), G88A ($n = 26$), G88R ($n = 27$), G88E ($n = 29$), or G88K ($n = 28$) mutant TASK-4 cRNA subjected to a 1-s test pulse to +40 mV (protocol as described above). Mean current amplitude at +40 mV was plotted relative to the currents generated by wild-type TASK-4 channels.

Data are provided as mean ± SEM. P values calculated in unpaired Student's *t*-test are indicated. Numbers of independent experiments are indicated within the bars.

acidic residue (glutamate) had a comparable effect as the substitution to a basic residue (lysine and arginine). However, while glycine mutations to arginine (R), lysine (K), and glutamate (E) resulted in a gain-of-function, introducing a small alanine residue (G88A) leads only to a small and not significant current increase (Fig 5C). Thus, our data indicate that a small residue (e.g., glycine or alanine) at position 88 is essential for normal TASK-4 channel gating.

## G88R mutant channel subunits cause a dominant-active gain-of-function

Since the patient was heterozygous for the G88R mutation, we co-expressed wild-type TASK-4 with the mutant channel to address the question whether there is also a gain-of-function present in the heterozygous state. For this purpose, we injected 25 ng of cRNA encoding wild-type TASK-4 into *Xenopus* oocytes or 12.5 ng to mimic a haploinsufficiency (Fig 6A and B). All data were normalized to the injections with 25 ng of TASK-4. G88R mutant channels were expressed alone (25 ng) or co-injected with cRNA of TASK-4 plus G88R (12.5 ng each) to mimic the heterozygous state (Fig 6A and B). The relative current after injection of G88R alone was increased by a factor of 2.58 ± 0.14 ($n = 33$), and G88R mutants co-expressed with TASK-4 had a relative current amplitude increase of 2.17 ± 0.16 ($n = 17$) (Fig 6A and B). Reflecting the observed heterozygosity in the patient, heteromeric channels formed by wild-type and mutant subunits had a strong gain-of-function. The strong gain-of-function observed for the co-expression suggests that G88R might act in a dominant-active manner (Supplementary Fig S3). Two-pore domain potassium channels are dimeric;

thus, for a co-expression, one would expect that 83.3% of the assembled channels contain at least one G88R subunit and 16.67% harbor TASK-4 wild-type subunits only (Fig 6C). For a dominant-active effect of G88R, the current amplitude after co-injection with wild-type channels would be composed of 83.3% of the G88R current amplitude (also reflected by the injection of 20.83 ng G88R cRNA) plus 16.67% of the current amplitude after injection of TASK-4 wild-type subunits only. The most straightforward interpretation of our data is, assuming a regular assembly, that the G88R mutant acts in a dominant-active manner (Fig 6B). The dominant-active gain-of-function by the G88R exchange suggests that in heterozygous patients, the majority of native cardiac TASK-4 channels should have the observed gain-of-function phenotype.

## G88R mutants stabilize the membrane potential and slow upstroke velocity of spontaneously beating HL-1 cells

Since TASK-4 is not expressed in mice, it is not possible to develop a transgenic 'G88R mouse' as a disease model for PCCD. As we found that TASK-4 is preferentially expressed in the conduction system, transfection of G88R into ventricular cardiomyocytes would not provide sufficient mechanistic information to explain the effects of the mutation on conductivity. HL-1 cells are spontaneously beating sinoatrial node like cardiomyocytes (Claycomb *et al*, 1998), and as these are more closely related to cells in the conduction system, we performed additional experiments using this cell type. We transfected EGFP-tagged wild-type TASK-4 or G88R in HL-1 cells and measured action potential frequency of the spontaneously beating HL-1 cells (Fig 7A and Supplementary Movies S1–S3) and character-

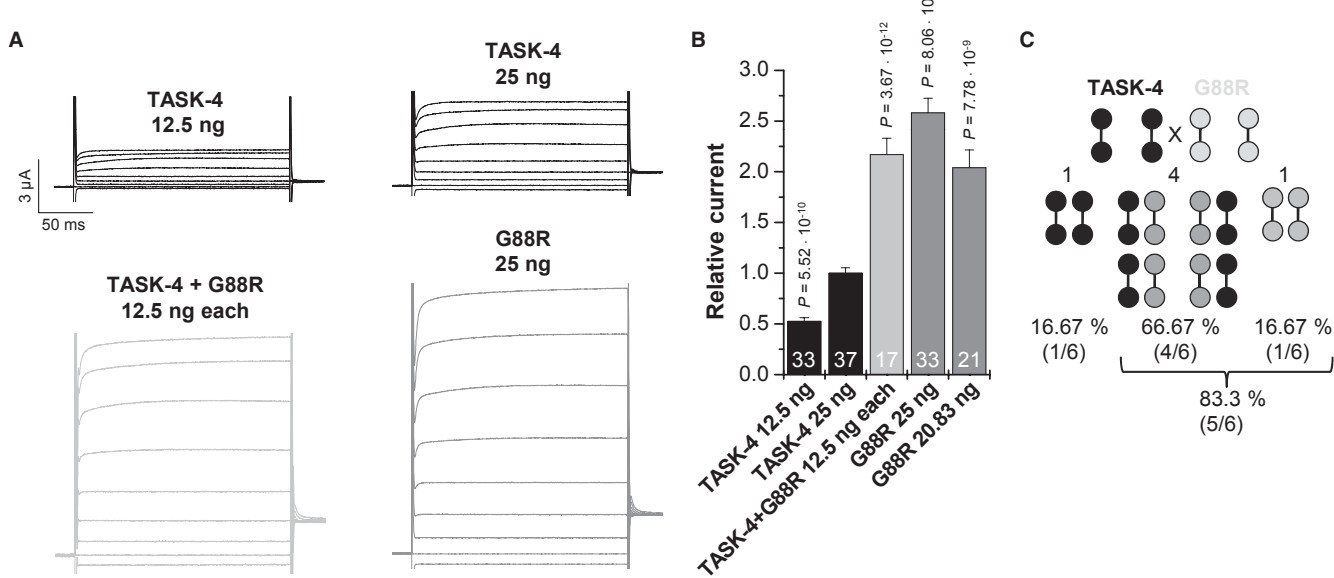

**Figure 6.   G88R mutant TASK-4 channel subunits have a dominant-active effect on wild-type TASK-4 subunits.**

A    Representative current voltage relationship measurements in *Xenopus* oocytes injected with wild-type TASK-4 cRNA (25 ng/oocyte, black), with 12.5 ng wild-type TASK-4 (mimicking haploinsufficiency), with wild-type TASK-4 and G88R-TASK-4 cRNA (12.5 ng each/oocyte, light gray) mimicking an heterozygous state or G88R mutant cRNA alone (25 ng/oocyte, gray). Voltage was stepped from a holding potential of −80 mV to potentials ranging from −100 mV to +60 mV in 20 mV increments. Voltage step duration was 200 ms and voltage steps were repeated every 2 s. Test pulses were followed by a step to −40 mV.

B    Mean current amplitude at 0 mV is blotted relative to TASK-4 wild-type currents (25 ng). Relative TASK-4 current after injection of 20.83 ng G88R-TASK-4 cRNA is thought to resemble the theoretical amount of dimeric channels containing at least one G88R subunit (83.3%). Data are provided as mean ± SEM. P values calculated in unpaired Student's *t*-test are indicated. Numbers of independent experiments are indicated within the bars.

C    Theoretical probability of dimeric subunit assembly after injection of similar amounts of wild-type and G88R-TASK-4 channel subunits.

ized the action potentials using patch clamp experiments (Fig 7C–J). Transfection of wild-type TASK-4 into HL-1 cells already slowed the action potential frequency from 179 ± 4 bpm to 125 ± 2 bpm (Fig 7A and Supplementary Movies S1 and S2), as expected for the overexpression of a tandem K⁺ channel in cells with less hyperpolarized membrane potentials, as in the sinoatrial node or in the conduction system. Most importantly, transfecting the same amount of G88R TASK-4 cDNA, with a similar efficiency and similar protein expression (Fig 7B), caused a significantly more pronounced slowing of the spontaneous beating frequency (Fig 7A and Supplementary Movie S2), and the frequency was reduced to 59 ± 3 bpm. In patch clamp recordings, the action potential frequency of untransfected HL-1 cells was much slower (Fig 7D), presumably reflecting the lack of supplemented Claycomb media which for instance contains norepinephrine. However, even under these non-stimulated conditions, the action potential frequency, recorded in the current-clamp mode, of G88R transfected cells was much slower than that of TASK-4 transfected cells (Fig 7C and D). In addition, the patch clamp experiments showed that the overexpression of G88R, compared to TASK-4, leads to a significantly more pronounced shortening of the action potential duration (Fig 7C and E), while the maximal diastolic membrane potential is more hyperpolarized (Fig 7C). This effect by G88R can be quantified by a more pronounced after hyperpolarization following the action potential (Fig 7F and G). Overexpression of TASK-4 and G88R also antagonizes depolarization, which can be noted by a reduced action potential overshoot (Fig 7H and I) and a strong slowing of the

upstroke velocity (Fig 7H and J). While the reduction of the action potential overshoot was already fully achieved by the overexpression of wild-type TASK-4 (Fig 7H and I), the gain-of-function by G88R caused a much more pronounced slowing of the upstroke velocity (Fig 7H and J). In summary, these overexpression experiments demonstrate that G88R leads to similar, but much stronger effects than the overexpression of wild-type TASK-4. Our data indicate that wild-type TASK-4 can hyperpolarize the resting membrane potential of cells in the conduction system and that the G88R mutation might hinder or slow the upstroke of an action potential by requiring a strong depolarization to overcome the leak channel's effect.

Thus, we propose that a stabilization of the membrane potential in the conduction system by G88R and especially a slowed upstroke velocity in the conduction system might contribute to the phenotype of slowed conductivity in PCCD.

## Discussion

In a patient with an unusual, severe phenotype combined of PCCD and IVF, we identified a digenic mutation state after WES. First, a putative loss-of-function mutation located within the cardiac sodium channel gene *SCN5A,* which is known to be associated with BrS, IVF, PCCD, SND, AF, and DCM (Abriel & Zaklyazminskaya, 2013), was identified. The essential splice site mutation identified at (+1 intron site, c.3963+1G>A) in *SCN5A* is novel, but

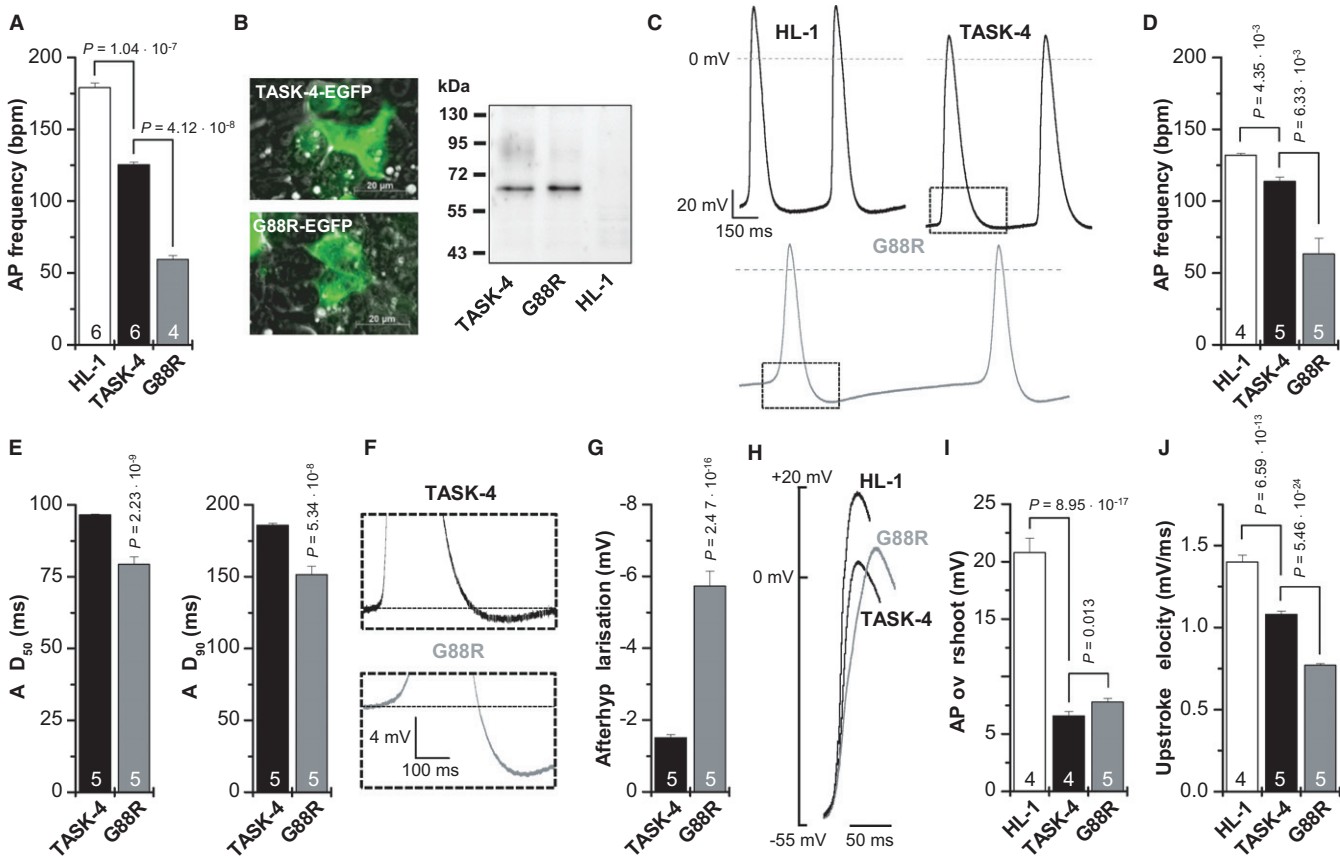

**Figure 7.  G88R mutants stabilize the membrane potential and slow upstroke velocity of spontaneously beating HL-1 cells.**

A   Action potential (AP) frequency, as beats per minute (bpm), of spontaneously beating HL-1 cells and of HL-1 cells transfected with TASK-4-EGFP or G88R-EGFP, was counted in supplemented Claycomb media. Beating frequency was determined from four to six independent transfections and dishes with untransfected HL-1 cells.

B   Fluorescence imaging and Western blot analyses of HL-1 cells transfected with TASK-4-EGFP or G88R-EGFP. For Western blot analysis, TASK-4 or G88R was detected with an anti-GFP antibody. HL-1, non-transfected HL-1 cells.

C   Patch clamp experiments in the current-clamp mode of HL-1 cells and HL-1 cells transfected with TASK-4-EGFP or G88R-EGFP. Boxes indicate the zoom area for panel F. The analyses of the patch clamp data in (D–J) were performed from four to five independent transfections for TASK-4 and G88R, respectively.

D   Action potential (AP) frequency, as beats per minute (bpm), of spontaneously beating HL-1 cells and of HL-1 cells transfected with TASK-4-EGFP or G88R-EGFP, recorded in the current-clamp mode under physiological saline conditions.

E   Analyses of the action potential duration, $APD_{50}$ and $APD_{90}$, of HL-1 cells transfected with TASK-4-EGFP or G88R-EGFP.

F   Illustration of the hyperpolarization observed following an action potential of HL-1 cells transfected with TASK-4-EGFP or G88R-EGFP.

G   Analyses of the afterhyperpolarization observed in HL-1 cells transfected with TASK-4-EGFP or G88R-EGFP.

H   Illustration of the action potential overshoot and the upstroke velocity of HL-1 cells and of HL-1 cells transfected with TASK-4-EGFP or G88R-EGFP. Note that the threshold for the action potentials was not significantly different for the constructs.

I   Analyses of the action potential (AP) overshoot of HL-1 cells and HL-1 cells transfected with TASK-4-EGFP or G88R-EGFP.

J   Analyses of the upstroke velocity (mV/ms) of HL-1 cells and HL-1 cells transfected with TASK-4-EGFP or G88R-EGFP.

Data are provided as mean ± SEM. *P* values calculated in unpaired Student's *t*-test are indicated. Numbers of independent experiments are indicated within the bars.

comparable to another mutation at the same donor splice site (+2 intron site, c.3963+2) that has been first reported in familial PCCD (Schott *et al*, 1999). Within the reported family, mutation carriers showed typical signs of PCCD (e.g., RBBB, LBBB, AVB, or hemiblock) and were subsequently treated with a pacemaker device. In contrast, IVF—as seen in the present case—has not been noted in any of these patients. Since there was no additional evidence for structural heart disease or external factors facilitating IVF, we hypothesized that our index case might be genetically different than the family members reported before (Schott *et al*, 1999). We therefore reasoned that the occurrence of IVF together with the severity of the cardiac conduction disturbance might be

due to a second (genetic or other) hit. WES is a reasonable strategy for discovering rare alleles which are responsible for Mendelian phenotypes and possibly for complex traits, too (Bamshad *et al*, 2011). For example, using WES calmodulin (*CALM1* and *CALM2*), mutations were discovered as the underlying genetic cause of severe QT-prolongation and recurrent cardiac arrest in infants (Crotti *et al*, 2013). In the present study, WES was utilized to identify the *KCNK17* gene encoding the $K_{2P}$ channel TASK-4 as a novel disease gene that might be functionally relevant for cardiac conduction disorders.

Prioritization of WES data in this approach revealed, in addition to the confirmation of the essential splice site mutation in *SCN5A*, a

functionally relevant variant in the ion channel gene *KCNK17* which likely modulates the electrical conduction in the heart of the patient with complex PCCD and IVF. Beginning with 76,712 variants with a reliable coverage identified in the exome of the patient, it is essential to carefully consider strategies for successfully filtering pathogenic variants. Prioritization schemes for severe Mendelian disorders should reflect that the mutation has strong consequences and, therefore, is novel or at least very rare in the general population, located within protein-coding regions and directly affecting protein function as a result of the gene mutation (Gilissen *et al*, 2012). Both remaining variants in *SCN5A* and *KCNK17* fulfilled these assumptions. However, it should be mentioned that filtering and gene prioritization steps might have excluded other putative pathogenic variants. Prioritization alone will not discover causal mutations, but in combination with common strategies like candidate gene approach and functional analysis, as it was applied in the current study, it might have success. In the current study, functional characterization of the *KCNK17* gene mutation was essential to prove that it may modulate the pathogenic impact on top of the *bona fide SCN5A* mutation. Although our experimental evidence supports the likelihood that TASK-4 is a new disease gene that is functionally relevant for cardiac conduction disorders, it remains formally possible that one of the other gene variants is also relevant or contributing to the phenotype. Even though the algorithm and PPT analysis predict no effect of those other variants, there is no direct experimental evidence to definitively rule out the role of other genes. As mice do not have a *KCNK17* gene and thus cannot be used to develop a disease model for TASK-4 mutations, there is no obvious way to 'prove' that the TASK-4 mutation is actually responsible in this individual—the evidence is, by nature, circumstantial. Nevertheless, the combination of our genetic data, the newly identified preferential expression of TASK-4 in the conduction system, together with the strong electrophysiological phenotype that we have identified, clearly suggests that the G88R mutation is a disease-modifying mutation for PCCD.

Mutations in *SCN5A* are known to be associated with a variety of cardiac phenotypes including LQT3, BrS1, PCCD and non-progressive CCD, AF, IVF, DCM, and SND (Abriel & Zaklyazminskaya, 2013). The reasons of this multiple allelic effects are only understood in part. However, the prediction of a phenotypic outcome (disease manifestation and clinical severity) is still of high relevance. This may resemble a mixture of one or more disease-related mutations and environmental or genetic modifiers (Crotti, 2011). In principle, a second mutant or polymorphic allele might exhibit also protective effects. This has been reported, for example, for the H558R polymorphism in the *SCN5A* gene. When the polymorphic R558 allele was present on the second allele (in trans), the loss-of-sodium channel function due to the trafficking-deficient R282H mutation could be restored *in vitro*. Moreover, family members with the *SCN5A* R282H mutation and the *SCN5A* R558 allele in trans were reported healthy, while the mutation carriers without a second polymorphic allele showed BrS (Poelzing *et al*, 2006). On the other site, interaction of multiple variants in different genes can provoke or aggravate a phenotype as well. Two common polymorphisms in regulatory sequences of the connexin40 (*GJA5*) gene have been reported to predispose to familial atrial standstill if they occur in conjunction with a *SCN5A* mutation, in contrast, carrying only a single genotype was relatively benign (Groenewegen *et al*, 2003). Thus, as already

shown for other excitability disorders such as LQTS, BrS, or familial epilepsy (Crotti *et al*, 2005; Poelzing *et al*, 2006; Klassen *et al*, 2011), phenotypic variation in cardiac arrhythmia may be originated in the functional interaction of mutations in known predisposing genes (*SCN5A*) with additional mutations in novel genes as shown here with *KCNK17*.

TASK-4 belongs to the group of $K_{2P}$ potassium channels, containing 15 members of leak channels (Decher *et al*, 2001; Girard *et al*, 2001). The current kinetics of $K_{2P}$ channels are relatively similar to those of the so-called 'plateau current' ($I_{K,P}$) observed in guinea-pig cardiomyocytes (Yue & Marban, 1988; Backx & Marban, 1993) or to the steady state $K^+$ current ($I_{K,SS}$) in mouse cardiomyocytes (Xu *et al*, 1999). Cardiomyocytes express several $K_{2P}$ channels, with TREK-1 and TASK-1 as the most studied cardiac leak channels (Yue & Marban, 1988; Xian Tao *et al*, 2006; Putzke *et al*, 2007; Decher *et al*, 2011; Limberg *et al*, 2011; Schiekel *et al*, 2013). We have provided the first quantitative description of the cardiac $I_{TASK-1}$ and determined the contribution of TASK-1 channels to the repolarization of rat (Putzke *et al*, 2007) and mouse (Decher *et al*, 2011) ventricular cardiomyocytes. TASK-1 channels seem to play a dominant role in the conduction system of the mouse which became evident from the preferential expression of the channel in this tissue and the QRS complex disturbances that can be observed in the TASK-1 knockout mouse (Graham *et al*, 2006; Decher *et al*, 2011; Donner *et al*, 2011). In addition, TASK-1 channels modulate action potential duration in human atrial cardiomyocytes (Limberg *et al*, 2011). As there are no specific TASK-4 blockers available and mice do not functionally express a *KCNK17* gene, little is known about the function of the pH-sensitive $K_{2P}$ channel TASK-4 in the heart (Decher *et al*, 2001; Girard *et al*, 2001). We showed that TASK-4 is, similar as TASK-1, preferentially expressed in the cardiac conduction system and less abundant in the ventricle. The G88R mutation is located in the extracellular loop between the first transmembrane segment and the first pore domain of the TASK-4 subunit. The amino acid exchange leads to a strong increase in TASK-4 conductance. A TASK-4 gain-of-function mutation is likely to promote repolarization of the cardiac action potential, which in turn might favor reentry arrhythmias due to a shortened effective refectory period. In addition, as TASK-4 is preferentially expressed in Purkinje fibers, the gain-of-function by G88R might hyperpolarize the membranes of cells in the conduction system and thus slow conductivity.

Gain-of-function mutations are rarely found in inherited forms of arrhythmias. So far, gain-of-function mutations in *KCNQ1*, *KCNH2*, and *KCNJ2* have been associated only with rare cases of short QT syndrome (SQTS) and/or AF. Only in some studies, co-expression experiments with wild-type channels (reflecting the identified heterozygosity in the patients) were performed, and here, ion currents showed an intermediate phenotype behavior (between the wild-type and mutant channel dysfunction). To our knowledge, a dominant-active phenotype—as observed for the TASK-4 G88R mutant in the present study—has not been described before. Thus, the *KCNK17* mutation is the first dominant-active arrhythmia-associated potassium channel mutation.

Currently, there are only three other $K_{2P}$ channel genes in which disease-causing mutations were identified. In a single report, familial Birk Barel syndrome that is characterized by mental retardation, hypotonia, and facial and skeletal dysmorphism is caused by a

mutation in the paternally imprinted potassium channel TASK-3 (*KCNK9*) on chromosome 8q24.3 (Barel *et al*, 2008). The non-synonymous mutation identified in the fourth transmembrane segment of TASK-3 (G236R) is expected to lie within the ion-conduction pathway of the channel and showed a dominant-negative effect on both, TASK-3 and TASK-1 channels. In addition, familial migraine with aura was linked to a single, dominant-negative mutation in the TRESK potassium channel gene (*KCNK18*) on chromosome 10q25.3 (Lafreniere *et al*, 2010). The 2-bp deletion (p.Phe139Trp fs*25) causes a frameshift, and the prematurely truncated protein resulted in a complete loss-of-channel function. However, not all non-synonymous *KCNK18* variants linked to migraine exhibited a clear loss-of-function as reported before. Thus, the genetic background of this disorder is more complex, and the complete genetic information of a patient may be necessary to understand the mechanisms for disease penetrance (Andres-Enguix *et al*, 2012). In addition, just recently, multiple mutations in the acid-sensitive $K_{2P}$ channel TASK-1 (*KCNK3*) were reported to cause familial and idiopathic pulmonary arterial hypertension (Ma *et al*, 2013). Thus, $K_{2P}$ channels appear to be a novel and potentially emerging ion channel class for various diseases. In this line, using WES and functional studies, we now link the first cardiac disorder (PCCD) to the $K_{2P}$ channel TASK-4.

# Materials and Methods

### Subject ascertainment and phenotypic analysis

A total of 463 patients with various arrhythmia syndromes were comprised in this study (AF, *n* = 10; AVB, *n* = 20; BrS, *n* = 200; IVF, *n* = 125; PCCD, *n* = 49; RVOT, *n* = 35; SND, *n* = 24) and finally completely sequenced for the presence of a *KCNK17* gene mutation. Comprehensive phenotypic analyses in the patient populations included history, physical examination, and ECG. In addition, in some patients, coronary angiography and cardiac magnetic resonance imaging were performed. The control cohort comprises 379 unrelated, healthy Caucasians.

### Ethical approval

Genotyping was approved by the ethic committee of the 'Ärztekammer Westfalen-Lippe' and the 'Medizinische Fakultät der Westfälischen Wilhelms-Universität' and conformed to the principles set out in the WMA Declaration of Helsinki and the NIH Belmont Report. Written informed consent was obtained from all individuals and patients. We did not have an explicit consent for depositing the clinical data in the dbGAP and EGA repositories.

### Enrichment, library preparation, and sequencing for WES

Patient genomic DNA, isolated from whole blood, was delivered for exome sequencing. In brief, the sample was prepared and enrichment was carried out according to Agilent's Sure Select Protocol Version 1.2. Concentration of the library was determined using Agilent's QPCR NGS Library Quantification Kit and adjusted to final concentration of 10 nM. Sequencing was performed on the Illumina HiSeq2000 platform using TruSeq v3 chemistry.

### Read mapping and alignment

Read files (FastQ) were generated from the sequencing platform via the manufacturer's proprietary software. Reads were mapped to their location in the most recent build of the human genome (hg19/b37) using Burrows-Wheeler Aligner (BWA) package (version 0.6.1). Local realignment of the mapped reads around potential insertion/deletion (indel) sites was carried out with the Genome Analysis Tool Kit (GATK) version 1.4. This algorithm ensures that the alignment has the minimum number of mismatching bases across the reads. The biggest effect of this is to reduce false-positive SNP calls around indels, and accurately determine indel length. Duplicate reads were marked using Picard version 1.62. Additional BAM file manipulations were performed with Samtools 0.1.18. Base quality (Phred scale) scores were recalibrated using GATK's covariance recalibration. This improves the accuracy of the base quality metrics which in turn improves the quality of variant calls. SNP and indel variants were called using the GATK Unified Genotyper. Variants were annotated using Ensembl to show which genes and transcripts are affected, and whether a variant is *a priori* likely to have a serious functional consequence.

### Nucleotide variant analysis

In light of expected non-synonymous variants in all genes, a prioritization scheme was developed to systematically identify candidate genes for a cardiac conduction disease with IVF. First, all variants with a low confidence (coverage <20-fold) were not followed. Next, variants not being present in 388 prioritized genes relevant for cardiac function (CARDIO gene panel; E. Schulze-Bahr, personal information) were excluded. Within this group, all non-genic, intronic, and synonymous variants were removed. Only alterations with serious consequences, namely non-synonymous coding, essential splice site, frameshift coding, stop gained or stop lost, or complex indel variations were further taken into account. Hereafter, all variants annotated in Exome Variant Server at the NHLBI (EVS, http://evs.gs.washington.edu/EVS/), dbSNP137 (www.ncbi.nlm.nih.gov/projects/SNP/), and Ensembl Gene Browser (http://www.ensembl.org) (MAF ≥ 0.01%) were excluded, and remaining variants confirmed by direct, bidirectional Sanger sequencing in two independent reactions.

To exclude that potentially disease-causing variants were ignored, it was checked whether variants with a non-reliable coverage <20-fold which were novel and had serious consequences were among the group of 388 prioritized genes and no variants were detected.

### Pathogenicity prediction tools

The effect of validated amino acid substitutions was predicted by PolyPhen-2 (Adzhubei *et al*, 2010), MutPred (Li *et al*, 2009), SNAP (Bromberg & Rost, 2007), SNPs&Go (Calabrese *et al*, 2009), and SIFT/Provean (Kumar *et al*, 2009; Choi *et al*, 2012). Each algorithm predicts pathogenicity upon specific parameters including sequence conservation and known functional motifs, and their use has been proposed as a method for classifying variants identified by re-sequencing (Hindorff *et al*, 2009). Non-synonymous single nucleotide variants (nsSNV) which were concordantly predicted as tolerated were denoted as non-synonymous SNP, and variants with a

discrepant prediction between the programs were subsequently classified as variant of unknown significance (VUS). All other variants with a concordantly predicted deleterious impact were further analyzed. For in-frame insertions and deletions, SIFT/Provean was used. Furthermore, all variants were annotated with Alamut version 2.2 (Interactive Biosoftware).

### Genotyping

Genomic DNA was extracted from whole blood, and *KCNK17* (Gen-Bank accession number AF358910.1) was completely sequenced in 463 patients with various arrhythmia syndromes as described before (Schulze-Bahr *et al*, 2003). Exon 2 of *KCNK17* comprising the identified nucleotide was sequenced in a control cohort of 379 unrelated, healthy individuals. Primer sequences and PCR conditions are available on request.

### Regional myocardial transcript detection

QuantiTect Reverse Transcription Kit (Qiagen) was used to transcribe cDNA from human total RNA from atrium, ventricle, SA- and AV-node as well as Purkinje fibers (Analytical Biological Services Inc.). Quantitative real-time PCR of *KCNK17* was performed three to seven times in triplicate using Rotor-Gene SYBR Green PCR Kit and Rotor-Gene Q (Qiagen) in comparison with *KCNK3*. For normalization, $\Delta Ct$-values for *KCNK17* and *KCNK3* were determined against *GAPDH* and averaged. The $\Delta Ct$ means for *KCNK17* in each heart compartment were used to describe relative mRNA expression normalized to ventricular *KCNK17* expression applying $2^{-\Delta\Delta Ct}$-method.

### Molecular biology

QuikChange Site-Directed Mutagenesis Kit (Stratagene) was used to introduce mutations into human TASK-4 (*KCNK17*) cDNA. For the chemiluminescence assay, a hemagglutinin (HA)-tag (YPYDVPDYA) was introduced at amino acid position 234 of human TASK-4. The constructs for oocyte recordings and the chemiluminescence assay were cloned into a pSGEM vector, and cRNA was prepared with mMESSAGE mMACHINE T7 Kit (Ambion) after linearization with NheI. For fluorescence imaging, TASK-4 constructs were subcloned into pDsRed-Monomer or pEGFP-vectors (Clontech).

### Heterologous expression of TASK-4 channels in *Xenopus* oocytes

Ovarian lobes were dissected from mature *Xenopus laevis* anesthetized with tricaine and treated with collagenase (1 mg/ml, Worthington, type II) in OR2 solution (in mM: NaCl 82.5, KCl 2, MgCl$_2$ 1, HEPES 5, pH 7.4) for 120 min. Isolated oocytes were stored at 18°C in ND96 recording solution (in mM: NaCl 96, KCl 2, CaCl$_2$ 1.8, MgCl$_2$ 1, HEPES 5) supplemented with Na-pyruvate (275 mg/l), theophylline (90 mg/l), and gentamicin (50 mg/l), pH 7.4. Stages IV and V oocytes were injected with TASK-4 cRNA (12.5, 20.85, or 25 ng). Standard two-microelectrode voltage-clamp recordings were performed at room temperature (21–22°C) 2 days after injection of oocytes with cRNA. Microelectrodes were fabricated from glass pipettes filled with 3 M KCl and had a resistance of 0.2–1.0 MΩ. Voltage-clamp recordings were performed at pH 8.5 with an

Axoclamp 900A amplifier, a Digidata 1440A and pClamp10 software (Axon Instruments). The holding potential was always −80 mV.

### Animals

The investigation conforms to the guide for the Care and Use of Laboratory Animals (NIH Publication 85-23). For this study, five female *Xenopus laevis* animals were used to isolate oocytes. Experiments using *Xenopus* toads were approved by the local ethics commission of the 'Regierungspräsidium Giessen'.

### Chemiluminescence assay

Surface expression of HA-tagged TASK-4 channel constructs was analyzed in *Xenopus* oocytes. Two days after cRNA injection, oocytes were incubated for 30 min in ND96 solution containing 1% bovine serum albumin (BSA) at 4°C to block non-specific binding of antibodies. Subsequently, oocytes were incubated for 60 min at 4°C with 1 μg/μl rat monoclonal anti-HA antibody (clone 3F10, Roche) in 1% BSA/ND96, washed six times at 4°C with 1% BSA/ND96, and incubated with 2 μg/μl peroxidase-conjugated affinity purified F(ab)2 fragment goat anti-rat IgG antibody (Dianova) in 1% BSA/ND96 for 60 min. Oocytes were washed thoroughly, initially in 1% BSA/ND96 (at 4°C for 60 min) and then in ND96 without BSA (at 4°C for 15 min). Individual oocytes were placed in 20 μl SuperSignal Elisa Femto solution (Pierce), and chemiluminescence was quantitated in a luminometer (Promega). The luminescence produced by non-injected oocytes was used as a reference signal (negative control).

### Fluorescence imaging

HeLa cells were grown to about 50% confluence on 35-mm glass-bottom Petri dishes (Wellco) supplemented with Dulbecco's Modified Eagle Medium containing 10% fetal bovine serum (Invitrogen). Twenty-four hour later, the cells were transfected with EGFP-tagged TASK-4 constructs using Fugene 6 (Roche). The cells were cultured at 37°C for 12–72 h in an incubator supplied with 5% CO$_2$/95% air. Microscopy and imaging were performed with a Zeiss Axio Observer.Z1 microscope equipped with a Zeiss Plan-Apochromat 60×/1.40 Oil DIC objective and a standard Zeiss filter set for EGFP (38HE). For DsRed-tagged constructs, a filter set of AHF Analysentechnik AG (Tübingen, Germany) (F46-005) was used. Images were taken with a Zeiss 12bit 'AxioCam MRm' camera, and digital images were processed using Zeiss AxioVision Software. Digital unmixing was performed before analyzing co-localization.

### Cell culture, transfection, and Western blot analysis

HL-1 cells were grown in supplemented Claycomb medium at 37°C in an incubator supplied with 5% CO$_2$ as previously described (Claycomb *et al*, 1998). Cells were split in a 1:3 ratio when they showed 100% confluence and the ability to beat. For transfection, HL-1 cells were seeded on 35-mm dishes (Nunc) in a 1:2 ratio. Twenty-four hour after seeding, cells were transfected with either 1.5 μg of TASK-4-EGFP or G88R-TASK-4-EGFP constructs using Lipofectamine 2000 following manufacturer instructions

**The paper explained**

**Problem**

Progressive cardiac conduction disorder (PCCD) is a common arrhythmia with an onset of disease primarily in the fifth decade of life. While PCCD is one of the main causes of pacemaker implantations, the genetic basis for PCCD mostly remains elusive, and only in 20–30% of probands, the disease-causing mutations are identified. In our study, we investigated a severe case of PCCD combined with IVF, while the patient had a lack of exonic mutations in the 'classical' genes known to cause cardiac arrhythmias.

**Result**

Although, we identified an intronic splice site mutation in the sodium channel gene *SCN5*A, the genetic reason for the severe phenotype was still unclear. Thus, we performed WES and identified an additional mutation in the *KCNK17* gene encoding the $K_{2P}$ potassium channel TASK-4. The heterozygous change (c.262G>A) resulted in the p.Gly88Arg mutation in the first extracellular pore loop. Mutant TASK-4 channels generated threefold increased currents, acting in a dominant-active manner by an increased conductivity. We demonstrate that Gly88 is a crucial residue for normal TASK-4 gating behavior and that the channel is strongly expressed in the cardiac conduction system.

**Impact**

The cardiac expression pattern, together with the observed gain-of-function, explains why the identified TASK-4 mutation should contribute to the severe phenotype observed in our particular patient. Thus, TASK-4 channels are novel genes involved in PCCD, a disease with widely unknown genetic origins. Recently, the leak ($K_{2P}$) channels unexpectedly evolved as a novel family of channelopathy genes. Here, WES supports a second hit-hypothesis in cases of severe arrhythmias and identified *KCNK17* as the first arrhythmia gene encoding a $K_{2P}$ channel.

(Invitrogen). After 24 h, transfection media were exchanged and cells were incubated for 4 h in supplemented Claycomb media. Subsequently, the 100% confluent and beating cells were used in patch clamp and imaging experiments. For Western blot analysis, HL-1 cells were harvested 24 h after transfection, washed in PBS, and lysed 30 min on ice using 200 µl RIPA buffer (in mM: 50 TRIS-Base, 150 NaCl, 1% NP40, 0.25% sodium-desoxycholat, 1 EDTA, pH 7.4) supplemented with 10 µl protease inhibitor cocktail (Sigma). Insoluble material was separated by centrifugation (13,000 rpm for 15 min at 4°C). Supernatants were mixed with 2× SDS sample buffer, heat denatured at 95°C for 5 min, centrifuged at 5,000 rpm for 3 min, and separated on 10% SDS-polyacrylamide gels. Bradford assays were performed to ensure loading of equal total protein amounts. Protein was visualized by immunoblotting with mouse anti-GFP antibodies (clone GF28R, 1:2,000, Dianova). The binding of the primary antibodies was detected using peroxidase-conjugated goat anti-mouse IgG antibodies (sc-2005, 1:2,000, Santa Cruz Biotechnology) and a chemiluminescent extended-duration substrate (Super Signal West Dura, Thermo Scientific).

**Imaging and movies of HL-1 cells**

Fluorescence images and bright-field movies were performed with a Zeiss Axio Observer.Z1 as described above, using a Zeiss LD A-Plan 40×/0.05 Ph2 objective and 35-mm dishes (Nunc) without oil immersion. Movies were taken by Axio Vision LE Modul 'Digital High Speed Recorder' with an exposure time of 70 ms per picture, resulting in a 15-s movie with 14 frames per second. To determine the action potential frequency, the beats per minute (bpm) were visually analyzed by four different persons, who simultaneously counted the beating rate in order to determine the average bpm. The average bpm was analyzed from four to six independent transfections and dishes with untransfected HL-1 cells.

**Patch clamp experiments of HL-1 cells**

HL-1 cells were transfected with either TASK-4-EGFP or G88R-TASK-4-EGFP constructs as described above. Action potentials were recorded in the whole-cell configuration under current-clamp conditions at room temperature (22°C). For patch clamp experiments, HL-1 cells were superfused with solution containing (in mM) 140 NaCl, 5.4 KCl, 1 $CaCl_2$, 1 $MgCl_2$, 0.33 $NaH_2PO_4$, 10 glucose, and 5 HEPES (pH 7.4 with NaOH) as previously described (Putzke *et al*, 2007). Patch clamp experiments were performed in the whole-cell configuration using pipettes pulled from borosilicate glass capillaries. The pipettes had a tip resistance of 3.0–4.0 MΩ when filled with a solution containing (in mM): 60 KCl, 65 K-glutamate, 5 EGTA, 3.5 $MgCl_2$, 2 $CaCl_2$, 3 $K_2ATP$, 0.2 $Na_2GTP$, and 5 HEPES (pH 7.2 with KOH). Data acquisition and command potentials were controlled with a commercial software program (Patchmaster, HEKA) with a sweep time interval of 1 s and a sample rate of 200 kHz. Data analysis of action potentials was done using the Fitmaster software (HEKA). For each cell measured, the action potential parameters were averaged by analyzing ten subsequent action potentials.

**Data analyses**

Results are reported as mean ± SEM (*n*, number of independent experiments). Statistical differences were evaluated by Student's unpaired *t*-test. *P* values are provided within the respective figure.

**Supplementary information** for this article is available online: http://embomolmed.embopress.org

## Acknowledgements

We thank Oxana Nowak, Sabine von Rueden, and Martina Raetz for excellent technical assistance and Prof. W.C. Claycomb for HL-1 cells. This work was supported by grants of the Deutsche Forschungsgemeinschaft (DE-1482/3-1 and 1482/3-2 to N.D., Schu1082/4-1 and 4-2 to E.S.-B.), the Anneliese Pohl-Habilitationsförderung der Anneliese Pohl-Stiftung to S.R. and the Fondation Leducq Paris (to E.S.-B.).

## Author contributions

CF analyzed the WES data and performed QPCR experiments. SR performed all functional recordings and cloned all constructs. SR and MFN performed the fluorescence microscopy. SZ and ESB assessed the clinical data. BS did screenings for genetic alterations. CF, SR, SZ, BS, and ND analyzed the data and made figures. ND and ESB designed the study. NS and AKK performed all experiments and data analyses with HL-1 cells. MFN, SZ, and BS helped rewriting the manuscript. CF, SR, ND, and ESB wrote the manuscript.

## Conflict of interest

The authors declare that they have no conflict of interest.

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
