## [Review Process File · EMBO Molecular Medicine]

Gain-of-function mutation in TASK-4 channels and severe cardiac conduction disorder

Corinna Friedrich, Susanne Rinné, Sven Zumhagen, Aytug K. Kiper, Nicole Silbernagel, Michael F. Netter, Birgit Stallmeyer, Eric Schulze-Bahr, Niels Decher

Corresponding author: Niels Decher, Philipps-University Marburg

Review timeline:

Submission date:	16 December 2013
Editorial Decision:	26 January 2014
Revision received:	24 April 2014
Editorial Decision:	08 May 2014
Revision received:	21 May 2014
Accepted:	21 May 2014

Transaction Report:

Editor: Céline Carret

1st Editorial Decision

26 January 2014

Thank you for the submission of your manuscript to EMBO Molecular Medicine and please accept my apologies for the delay (review process is always delayed across the Holiday period). We have now heard back from the three referees whom we asked to evaluate your manuscript. Although the referees find the study to be of potential interest, they also raise a number of concerns that have to be addressed in the next final version of your article.

As you will see from the comments below, all three referees are concerned about the conclusiveness of the data. Referee 1, while succinct, makes a pertinent point and suggests adding some family history to the study, which to our view would not only increase the conclusiveness but also the clinical significance of the findings. Referees 2 and 3 both mention that the SNP in Task4 is circumstantial to the phenotypic changes, but is not shown to be responsible for it and referee 3 suggests providing additional cellular data to strengthen this point.

Given the balance of these evaluations, we feel that we can consider a revision of your manuscript if you can address all issues that have been raised. Please note that it is EMBO Molecular Medicine policy to allow only a single round of revision and that, as acceptance or rejection of the manuscript will depend on another round of review, your responses should be as complete as possible.

***** Reviewer's comments *****

Referee #1 (Remarks):

A case is made at the level of plausibility for an association of a TASK4 missense mutation with the PCCD/IVF phenotype in a single 63-year old patient. The TASK4 mutation confers gain of function of a plateau-type K current in oocytes. Together with a splice site variant in SCN5a which is predicted to cause loss of function, the authors argue that the genotype may explain the phenotype.

The case is plausible. It would have been further tested had any relatives been phenotyped and genotyped, but no family history is mentioned. Such information would substantially strengthen the manuscript.

Referee #2 (Remarks):

In this manuscript, the authors applied whole exome sequencing (WES) with a prioritization algorithm for recognizing disease-causing mutations, to identify a new channel mutation that appears to contribute to a severe cardiac abnormality in a human patient (PCCD: progressive cardiac conduction disorder; and IVF: idiopathic ventricular fibrillation). Thus, in addition to a mutation in the SCN5A Na channel gene, they also find a glycine-to-arginine substitution in KCNK17 that encodes the TASK-4 background K channel. Expression of TASK-4 was prominent in human cardiac conduction tissue and, in heterologous expression studies, the G88A mutation caused a strong gain-of-function in K current. Based on this evidence, the authors conclude that these mutations together conspire to yield the pronounced cardiac phenotype in this patient.

Overall, this a well-written and well-illustrated paper that provides new information supporting the idea that TASK-4 mutations can contribute to human arrhythmias. It also suggests a previously unappreciated role for TASK-4 in cardiac conduction systems. Finally, and also importantly, it presents a rational blueprint for using WES to identify novel disease-causing mutations in individual patients.

Minor Concerns:

1. The authors should qualify their conclusions somewhat. For example, in the Discussion, they state: page 16, "to prove that it modifies the pathogenic impact on top of the bona fide SCN5A mutation"; and page 15, "TASK-4 as a new disease gene that is functionally relevant for cardiac conduction disorders." Although the evidence and arguments strongly support this likelihood, it remains formally possible that one of the other gene variants is actually responsible. That is, even though the algorithm and PPT analysis predicts no effect of those other variants, there is no direct experimental evidence to rule them out definitively. Also, there is no obvious way to "prove" that the TASK-4 mutation is actually responsible in this individual - the evidence is, by nature, circumstantial.
2. It is not entirely certain that the currents generated from co-expression experiments reflect heterodimeric channels containing both a wild type and mutated subunit, rather than additive effects of distinct populations of homomeric channels comprising only wild type or mutated subunits. The quantitative analysis that predicted the same current amplitude for 20.83 ng injection may be simply fortuitous. In any case, it is not necessarily crucial for the overall interpretation whether or not the individual channels are truly heteromeric, so perhaps this conclusion should be tempered as well (e.g., see p. 13, and elsewhere).
3. It was surprising that no data were presented for TASK-4(G88R) at 12.5 ng in Fig. 6B. Please include those data.
4. For the uninitiated reader, it would be helpful to provide some markup of the panels in Fig.1 that

could highlight the key measures on the ECG and how they changed over the 5 year observation period.

Referee #3 (Comments on Novelty/Model System):

The manuscript by Friedrich and colleagues report a combined candidate gene and whole-exome sequencing strategy to determine the genetic basis for a severe case of combined progressive cardiac conduction system disease (PCCD) and idiopathic ventricular fibrillation (IVF) in a single adult male. They discovered a novel splice site mutation in the cardiac sodium channel gene (SCN5A) and a gain-of-function nonsynonymous variant in a twin-pore potassium channel (TASK-4) with a plausible, but unproven, role in cardiac electrophysiology.

The study main focuses on the functional consequences of the TASK-4 variant, which provide convincing evidence of gain of function. However, the main conclusions of the paper implicate TASK-4 as a genetic modifier of the phenotype, a conclusion that is severely weakened by the lack of any genotype-phenotype correlation. An effort to determine the ECG phenotypes of first degree relatives and the associated genotypes at SCN5A and TASK-4 might reveal additional evidence supporting their main claim. Further experiments in a myocyte system to understand the cellular consequences of a gain-of-function in TASK-4 would also strengthen the paper.

Referee #3 (Remarks):

The manuscript by Friedrich and colleagues report a combined candidate gene and whole-exome sequencing strategy to determine the genetic basis for a severe case of combined progressive cardiac conduction system disease (PCCD) and idiopathic ventricular fibrillation (IVF) in a single adult male. They discovered a novel splice site mutation in the cardiac sodium channel gene (SCN5A) and a gain-of-function nonsynonymous variant in a twin-pore potassium channel (TASK-4) with a plausible, but unproven, role in cardiac electrophysiology. The study main focuses on the functional consequences of the TASK-4 variant, which provide convincing evidence of gain of function. However, the main conclusions of the paper implicate TASK-4 as a genetic modifier of the phenotype, a conclusion that is severely weakened by the lack of any genotype-phenotype correlation.

Major comments

1. Addition genotype-phenotype data are required to provide a convincing argument that TASK-4 modifies the trait. An effort to determine the ECG phenotypes of first degree relatives and the associated genotypes at SCN5A and TASK-4 is essential. It remains possible that TASK-4 is unrelated to the phenotype.
2. Additional evidence should be provided to support that a gain-of-function variant in TASK-4 will affect cardiomyocyte resting potential or some other cellular electrophysiological phenotype. Rather than hyperpolarize the resting membrane potential, this variant might hinder or slow the upstroke of an action potential by requiring a strong depolarization to overcome the leak channel's effect. This information would be extremely valuable for supporting the author's main conclusions.

Minor concerns

- a. Some references appear as numbers in the text (see page 15, end of first paragraph for example).
- b. Page 17, 3rd line from bottom - the parenthetical '= pseudogene' has an obscure meaning. Please explain what you mean.

Referee #1 (Remarks):

A case is made at the level of plausibility for an association of a TASK4 missense mutation with the PCCD/IVF phenotype in a single 63-year old patient. The TASK4 mutation confers gain of function of a plateau-type K current in oocytes. Together with a splice site variant in SCN5a which is predicted to cause loss of function, the authors argue that the genotype may explain the phenotype.

The case is plausible. It would have been further tested had any relatives been phenotyped and genotyped, but no family history is mentioned. Such information would substantially strengthen the manuscript.

Thank you for the positive comments and for reviewing our manuscript. We absolutely agree that having a genotype-phenotype correlation for the family would substantially strengthen the manuscript and support our experimental conclusions. We tried hard to convince other family members, in particular first degree relatives to participate in our study. However, they clearly and strictly refused to join. Still, we were able to reconstruct the family history, at least, and already introduced this into the original manuscript (page 7). Here we previously noted *“The family history was negative for sudden cardiac death or known inherited cardiac conditions”*. In addition, we now explicitly state on page 10: *“Since DNA from other family members was not available, we were not able to proof whether the identified genetic mutations in both genes were inherited or occurred as de-novo ones, however the family history was not further indicative for other arrhythmias or sudden cardiac death”*.

Nevertheless, our study is as we think, an excellent example that the novel technique of whole exome sequencing, combined with PPT predictions and electrophysiological recordings, can provide answers in single arrhythmia cases, where in former days classical genetics would have failed.

Referee #2 (Remarks):

In this manuscript, the authors applied whole exome sequencing (WES) with a prioritization algorithm for recognizing disease-causing mutations, to identify a new channel mutation that appears to contribute to a severe cardiac abnormality in a human patient (PCCD: progressive cardiac conduction disorder; and IVF: idiopathic ventricular fibrillation). Thus, in addition to a mutation in the SCN5A Na channel gene, they also find a glycine-to-arginine substitution in KCNK17 that encodes the TASK-4 background K channel. Expression of TASK-4 was prominent in human cardiac conduction tissue and, in heterologous expression studies, the G88A mutation caused a strong gain-of-function in K current. Based on this evidence, the authors conclude that these mutations together conspire to yield the pronounced cardiac phenotype in this patient.

Overall, this a well-written and well-illustrated paper that provides new information supporting the idea that TASK-4 mutations can contribute to human arrhythmias. It also suggests a previously unappreciated role for TASK-4 in cardiac conduction systems. Finally, and also importantly, it presents a rational blueprint for using WES to identify novel disease-causing mutations in individual patients.

Thank you for reviewing our manuscript and the very positive and useful comments.

Minor Concerns:

1. The authors should qualify their conclusions somewhat. For example, in the Discussion, they state: page 16, "to prove that it modifies the pathogenic impact on top of the bona fide SCN5A mutation"; and page 15, "TASK-4 as a new disease gene that is functionally relevant for cardiac conduction disorders." Although the evidence and arguments strongly support this likelihood, it remains formally possible that one of the other gene variants is actually responsible. That is, even though the algorithm and PPT analysis predicts no effect of those other variants, there is no direct experimental evidence to rule them out definitively. Also, there is no obvious way to "prove" that the TASK-4 mutation is actually responsible in this individual - the evidence is, by nature, circumstantial.

We agree that there is no direct experimental evidence to definitively rule out that other factors contribute to the phenotype and thus, de-emphasized some of our statements according to your suggestion. The statement on page 17 (former page 16) was changed to "to prove that it may modulate the pathogenic impact on top of the bona fide SCN5A mutation". Accordingly we de-emphasized our statements on page 16 (former page 15): "TASK-4 as a novel and potentially relevant disease gene that might be functionally relevant for cardiac conduction disorders."

In addition, we added a Discussion section on page 17, at the end of the first paragraph: "Although our experimental evidence strongly supports the likelihood that TASK-4 is a new disease gene that is functionally relevant for cardiac conduction disorders, it remains formally possible that one of the other gene variants is also relevant or contributing to the phenotype. Even though the algorithm and PPT analysis predicts no effect of those other variants, there is no direct experimental evidence to definitively rule out the role of other genes. As mice do not have a KCNK17 gene and thus cannot be used to develop a disease model for TASK-4 mutations, there is no obvious way to "prove" that the TASK-4 mutation is actually responsible in this individual - the evidence is, by nature, circumstantial. Nevertheless, the combination of our genetic data, the newly identified preferential expression of KCNK17 in the conduction system, together with the strong electrophysiological phenotype that we have identified, clearly suggests that the G88R mutation is a disease modifying mutation for PCCD."

2. It is not entirely certain that the currents generated from co-expression experiments reflect heterodimeric channels containing both a wild type and mutated subunit, rather than additive effects of distinct populations of homomeric channels comprising only wild type or mutated subunits. The quantitative analysis that predicted the same current amplitude for 20.83 ng injection may be simply fortuitous. In any case, it is not necessarily crucial for the overall interpretation whether or not the individual channels are truly heteromeric, so perhaps this conclusion should be tempered as well (e.g., see p. 13, and elsewhere).

Thank you for discussing this point. To further highlight that the gain-of-function is not just additive, we now include the data of the 12.5 ng G88R injection and provide a new Figure (Supplementary Figure 2 and Results page 13 (last line) and page 14 (first line)). In the novel Supplementary Figure 2, we highlight that the study was performed in a linear range, as injection of twice the amount of TASK-4 cRNA into *Xenopus* oocytes leads to a doubling of the current amplitude. Injection of 12.5 ng G88R leads to a 2.58 ± 0.2 fold increase in current amplitude compared to injection of 12.5 ng wild-type cRNA. This is a similar gain-of-function, as we have observed, when using 25 ng of cRNA for the constructs (Fig. 6B). Co-expression of 12.5 ng wild-type TASK-4 with 12.5 ng of G88R leads to a pronounced current increase, which is bigger than adding the amplitudes for both the individual constructs (Supplementary Figure 2; calculation no heteromers). Note that such an additive behavior would only occur if the channels would not form heteromers and express as separate homomeric channels. However, there is no evidence that the G88R should fail to form heteromeric channels with wild-type TASK-4, especially as our fluorescence imaging does not show a separate G88R population (Fig. 4D). Most importantly, after co-expression with wild-type and assuming a normal assembly, only 16.67 % of the channels would have two G88R subunits (Fig. 6C). Thus, if the gain-of-function would not be conferred to heteromeric channels with wild-type subunits, only 16.67 % of the dimeric channels would show a gain-of-function. The resulting current would be formed by 16.67 % of the G88R amplitude plus 83.33 % of wild-type amplitude. Calculating the expected current when only the channels with two G88R subunits have a gain-of-function (Supplementary Figure 2; calculation non-dominant) shows that the observed strong current increase by co-expression can only be explained if heteromeric channels of wild-type and G88R subunits also have a gain-of-function. Thus, it is for us the most straightforward interpretation that G88R is assembled with wild-type subunits which is conferring a gain-of-function to the heteromeric channel complex.

Nevertheless, as suggested we tempered our statements i.e. on page 14 “*Our data clearly showed that the G88R mutant acts in a dominant-manner*” was changed to “*The most straightforward interpretation of our data is, assuming a regular assembly that the G88R mutant acts in a dominant-manner*”; And the last sentence of the first paragraph on page 14 was changed to “*The dominant-active gain-of-function by the G88R exchange suggests that in heterozygous patients the majority of native cardiac TASK-4.....*”.

3. *It was surprising that no data were presented for TASK-4(G88R) at 12.5 ng in Fig. 6B. Please include those data.*

These data were initially not included, as the experiments were designed to mimic the most common clinical states, meaning wild-type (two healthy alleles), a haploinsufficiency (only one healthy allele), a heterozygous (one wild-type and one mutant allele) and a homocytotic state (two mutant alleles). Thus, we did initially not provide the current amplitudes for 12.5 ng of the G88R mutant, as it would reflect the more unlikely situation of a mutant allele in the presence of a haploinsufficiency. As we were working in a linear range, providing the current amplitudes of the 12.5 ng G88R cRNA injection did, to our initial opinion, not provide any additional information, as we just observed the similar gain-of-function (2.6-fold), as when recording the mutant with 25 ng (2.9-fold in Fig. 4B and 2.6-fold in Fig. 6B). However, in the context of the question that was raised above (comment 3), we now include this data in the novel Supplementary Figure 2 to highlight that the gain-of-function is conferred to the heteromeric channels and that the increased amplitudes are not caused by an additive effect.

4. *For the uninitiated reader, it would be helpful to provide some markup of the panels in Fig.1 that could highlight the key measures on the ECG and how they changed over the 5 year observation period.*

Thank you for this suggestion. In the revised manuscript, we have now included markups for the relevant segments in the ECGs.

Referee #3 (Comments on Novelty/Model System):

The manuscript by Friedrich and colleagues report a combined candidate gene and whole-exome sequencing strategy to determine the genetic basis for a severe case of combined progressive cardiac conduction system disease (PCCD) and idiopathic ventricular fibrillation (IVF) in a single adult male. They discovered a novel splice site mutation in the cardiac sodium channel gene (SCN5A) and a gain-of-function nonsynonymous variant in a twin-pore potassium channel (TASK-4) with a plausible, but unproven, role in cardiac electrophysiology.

The study main focuses on the functional consequences of the TASK-4 variant, which provide convincing evidence of gain of function. However, the main conclusions of the paper implicate TASK-4 as a genetic modifier of the phenotype, a conclusion that is severely weakened by the lack of any genotype-phenotype correlation. An effort to determine the ECG phenotypes of first degree relatives and the associated genotypes at SCN5A and TASK-4 might reveal additional evidence supporting their main claim. Further experiments in a myocyte system to understand the cellular consequences of a gain-of-function in TASK-4 would also strengthen the paper.

Referee #3 (Remarks):

The manuscript by Friedrich and colleagues report a combined candidate gene and whole-exome sequencing strategy to determine the genetic basis for a severe case of combined progressive cardiac conduction system disease (PCCD) and idiopathic ventricular fibrillation (IVF) in a single adult male. They discovered a novel splice site mutation in the cardiac sodium channel gene (SCN5A) and a gain-of-function nonsynonymous variant in a twin-pore potassium channel (TASK-4) with a plausible, but unproven, role in cardiac electrophysiology. The study main focuses on the functional consequences of the TASK-4 variant, which provide convincing evidence of gain of function. However, the main conclusions of the paper implicate TASK-4 as a genetic modifier of the phenotype, a conclusion that is severely weakened by the lack of any genotype-phenotype correlation.

We thank Referee #3 for these clear comments and for reviewing our manuscript.

Major comments

1. Addition genotype-phenotype data are required to provide a convincing argument that TASK-4 modifies the trait. An effort to determine the ECG phenotypes of first degree relatives and the associated genotypes at SCN5A and TASK-4 is essential. It remains possible that TASK-4 is unrelated to the phenotype.

We absolutely agree that having a genotype-phenotype correlation for the family would substantially strengthen the manuscript and support our experimental conclusions. As stated for Referee #1, we did a lot of personal efforts to convince other family members, in particular first degree relatives to participate in our study. However, the fate of the index patient closed the door for others to participate - probably in the light of anxiety and potential recurrence of cardiac events in relatives. However, we were able to reconstruct the family history, at least, and already introduced this into the original manuscript (page 7). Here we previously noted *"The family history was negative for sudden cardiac death or known inherited cardiac conditions"*. In addition, we now explicitly state on page 10: *"Since DNA from other family members was not available, we were not able to proof whether the identified genetic mutations in both genes were inherited or occurred as de-novo ones, however the family history was not further indicative for other arrhythmias or sudden cardiac death"*.

Nevertheless, our study is as we think, an excellent example that the novel technique of whole exome sequencing can provide answers in single arrhythmia cases, where in former days classical genetics would have failed. In the revised manuscript we now carefully discuss the problems arising by identifying a novel arrhythmia gene in a single index patient utilizing whole exome sequencing (page 17, end of the first paragraph): *"Although our experimental evidence strongly supports the likelihood that TASK-4 is a new disease gene that is functionally relevant for cardiac conduction disorders, it remains formally possible that one of the other gene variants is also relevant or contributing to the phenotype. Even though the algorithm and PPT analysis predicts no effect of those other variants,*

there is no direct experimental evidence to definitively rule out the role of other genes. As mice do not have a KCNK17 gene and thus cannot be used to develop a disease model for TASK-4 mutations, there is no obvious way to "prove" that the TASK-4 mutation is actually responsible in this individual - the evidence is, by nature, circumstantial. Nevertheless, the combination of our genetic data, the newly identified preferential expression of KCNK17 in the conduction system, together with the strong electrophysiological phenotype that we have identified, clearly suggests that the G88R mutation is a disease modifying mutation for PCCD."

2. Additional evidence should be provided to support that a gain-of-function variant in TASK-4 will affect cardiomyocyte resting potential or some other cellular electrophysiological phenotype. Rather than hyperpolarize the resting membrane potential, this variant might hinder or slow the upstroke of an action potential by requiring a strong depolarization to overcome the leak channel's effect. This information would be extremely valuable for supporting the author's main conclusions.

Since TASK-4 is not expressed in mice, it is not possible to develop a transgenic G88R mouse as a disease model for PCCD. As TASK-4 is preferentially expressed in the conduction system, transfection of G88R into ventricular cardiomyocytes would not provide the necessary mechanistic information to explain the effects of the mutation on conductivity. HL-1 cells are spontaneously beating sino-atrial node like cardiomyocytes. As these are more closely related to cells in the conduction system, we performed additional experiments using this cell type. As Referee 3 already anticipated, the gain-of-function by G88R, as compared to the overexpression of TASK-4 in HL-1 cells, does not only cause a hyperpolarization, but also antagonizes depolarization. This can be noted by a strong slowing of the upstroke velocity. In addition, G88R induces a shortening of the action potential duration, a hyperpolarization following the action potential and a long phase of diastolic depolarization, resulting in a reduced action potential frequency of the spontaneously beating HL-1 cells.

The novel data has been introduced in the Results section on page 14-15 of the revised manuscript, the new Figure 7 and Figure Legend, the Methods section and the Supporting Information Movies S1 to S3.

We agree that the requested mechanistic experiments were extremely valuable for supporting our main conclusions. As in the original version of the manuscript, we propose that a stabilization of the membrane potential in the conduction system might lead to a slowed conductivity, but now we also highlight that a slowed upstroke velocity in the conduction system might contribute to the phenotype of PCCD. In addition, we hope that the re-discussion of our data in our point-by-point response address the remaining concerns raised by Referee 3.

In the revised Results section we have also included a quote of the cellular effects you proposed above. The novel Results section now reads:

"G88R mutants stabilize the membrane potential and slow upstroke velocity of spontaneously beating HL-1 cells

Since TASK-4 is not expressed in mice, it is not possible to develop a transgenic 'G88R mouse' as a disease model for PCCD. As we found that TASK-4 is preferentially expressed in the conduction system, transfection of G88R into ventricular cardiomyocytes would not provide sufficient mechanistic information to explain the effects of the mutation on conductivity. HL-1 cells are spontaneously beating sino-atrial node like cardiomyocytes (Claycomb et al, 1998) and as these are more closely related to cells in the conduction system, we performed additional experiments using this cell type. We transfected EGFP-tagged wild-type TASK-4 or G88R in HL-1 cells and measured action potential frequency of the spontaneously beating HL-1 cells (Fig 7A and Supporting Information Movies S1 to S3) and characterized the action potentials using patch clamp experiments (Fig 7C-J). Transfection of wild-type TASK-4 into HL-1 cells already slowed the action potential frequency from 179 ± 4 bpm to 125 ± 2 bpm (Fig 7A and Supporting Information Movies S1 and S2), as expected for the overexpression of a tandem K^+ channel in cells with a less hyperpolarized membrane potentials, as in the sino-atrial node

or in the conduction system. Most importantly, transfecting the same amount of G88R TASK-4 cDNA, with a similar efficiency and similar protein expression (Fig 7B), caused a significantly more pronounced slowing of the spontaneous beating frequency (Fig 7A and Supporting Information Movies S2) and the frequency was reduced to 59 ± 3 bpm. In patch clamp recordings the action potential frequency of untransfected HL-1 cells was much slower (Fig 7D), presumably reflecting the lack of supplemented Claycomb media which for instance contains norepinephrine. However, even under these non-stimulated conditions, the action potential frequency, recorded in the current clamp mode, of G88R transfected cells was much slower than that of TASK-4 transfected cells (Fig 7C and D). In addition, the patch clamp experiments showed that the overexpression of G88R, compared to TASK-4, leads to a significantly more pronounced shortening of the action potential duration (Fig 7C and E), while the maximal diastolic membrane potential is more hyperpolarized (Fig 7C). This effect by G88R can be quantified by a more pronounced afterhyperpolarisation following the action potential (Fig 7F and G). Overexpression of TASK-4 and G88R also antagonizes depolarization, which can be noted by a reduced action potential overshoot (Fig 7H and I) and a strong slowing of the upstroke velocity (Fig 7H and J). While the reduction of the action potential overshoot was already fully achieved by the overexpression of wild-type TASK-4 (Fig 7H and I), the gain-of-function by G88R caused a much more pronounced slowing of the upstroke velocity (Fig 7H and J). In summary, these overexpression experiments demonstrate that G88R leads to similar, but much stronger effects than the overexpression of wild-type TASK-4. Our data indicate that wild-type TASK-4 can hyperpolarize the resting membrane potential of cells in the conduction system and that the G88R mutation might hinder or slow the upstroke of an action potential by requiring a strong depolarization to overcome the leak channel's effect.

Thus, we propose that a stabilization of the membrane potential in the conduction system by G88R and especially a slowed upstroke velocity in the conduction system might contribute to the phenotype of slowed conductivity in PCCD.”

Minor concerns

a. Some references appear as numbers in the text (see page 15, end of first paragraph for example).

Thank you. We included the reference (page 16 of the revised manuscript).

b. Page 17, 3rd line from bottom - the parenthetical '= pseudogene' has an obscure meaning. Please explain what you mean.

The term `pseudogene` on former page 17 was removed and we now state on page 19: “As there are no specific TASK-4 blockers available and mice do not functionally express a KCNK17 gene”.

2nd Editorial Decision

08 May 2014

Thank you for the submission of your revised manuscript to EMBO Molecular Medicine. We have now received the enclosed reports from the referees that were asked to re-assess it. As you will see the reviewers are now globally supportive and I am pleased to inform you that we will be able to accept your manuscript pending the following final amendments:

- Please carefully address the last minor comments of both reviewers

Please submit your revised manuscript within two weeks.

I look forward to receiving a new revised version of your manuscript.

***** Reviewer's comments *****

Referee #1 (Comments on Novelty/Model System):

Medium medical impact because this is a rare, likely unique situation. But, the utility of WES for single-case forensics is highlighted and this is generalizable

Referee #1 (Remarks):

The lack of family data is lamentable, but the totality of the evidence is intriguing (if still circumstantial).

One small but important matter should be addressed in the final revision: the authors now state that specimens from family members were not available, but, in the responses to the reviewers, they clearly had contacted several family members who had refused to cooperate. It would be more useful to state that all known family members were contacted and refused genetic testing. The authors should also show, as a supplemental figure, a pedigree. If the proband is the only one affected, do indicate it.

Referee #3 (Remarks):

The revised manuscript by Friedrich et al., is improved from the perspective of a plausible pathophysiological effect of the reported TASK-4 variant. The new experimental data in HL-1 cells is convincing and compelling. There are 3 minor concerns remaining:

1. The abstract should include mention of the new mechanistic data.
2. I suggest deleting the word 'strongly' on page 17 (first sentence of revised section)
3. Correct misspelling of word 'observed' in Fig 7G legend (p. 40)

2nd Revision - authors' response

21 May 2014

Referee #1 (Comments on Novelty/Model System):

Medium medical impact because this is a rare, likely unique situation. But, the utility of WES for single-case forensics is highlighted and this is generalizable

Referee #1 (Remarks):

The lack of family data is lamentable, but the totality of the evidence is intriguing (if still circumstantial).

One small but important matter should be addressed in the final revision: the authors now state that specimens from family members were not available, but, in the responses to the reviewers, they clearly had contacted several family members who had refused to cooperate. It would be more useful to state that all known family members were contacted and refused genetic testing. The authors should also show, as a supplemental figure, a pedigree. If the proband is the only one affected, do indicate it.

Thank you for your review and your input. We have included the statement as suggested. See on page 10, second paragraph: “All known family members were contacted and refused genetic testing”.

As suggested we have prepared a new Supplementary Fig. 1, with the pedigree of the family. This pedigree indicates that we only know about arrhythmias in the index patient and that from the other family members no DNA was available.

Referee #3 (Remarks):

The revised manuscript by Friedrich et al., is improved from the perspective of a plausible pathophysiological effect of the reported TASK-4 variant. The new experimental data in HL-1 cells is convincing and compelling. There are 3 minor concerns remaining:

1. The abstract should include mention of the new mechanistic data.

Thank you for this good suggestion. As the Abstract has a limit of 175 Words, we could only include a short statement, which reads: “...that overexpression of G88R leads to a hyperpolarization and strong slowing of the upstroke velocity of spontaneously beating HL-1 cells.” As the Abstract was in the previous version already at the word limit, we had to delete a sentence from the Abstract, reporting that introducing other residues at position 88 by site-directed mutagenesis, also results in a gain-of-function. This way, we can report about the new mechanistic data with HL-1 cells, without losing any essential information within the Abstract.

2. I suggest deleting the word 'strongly' on page 17(first sentence of revised section)

Done.

3. Correct misspelling of word 'observed' in Fig 7G Legend (p. 40)

Corrected. Thank you.